# A synthetic nanobody targeting RBD protects hamsters from SARS-CoV-2 infection

Tingting Li[1,12], Hongmin Cai[1,12], Hebang Yao[1,12], Bingjie Zhou[2,3,12], Ning Zhang[4,12], Martje Fentener van Vlissingen[5,6], Thijs Kuiken[6,7], Wenyu Han[1,2], Corine H. GeurtsvanKessel[6,7], Yuhuan Gong[2,4], Yapei Zhao[2,3], Quan Shen[4], Wenming Qin[8], Xiao-Xu Tian[8], Chao Peng[8], Yanling Lai[1,2], Yanxing Wang[1], Cedric A. J. Hutter[9], Shu-Ming Kuo[3], Juan Bao[1], Caixuan Liu[1,2], Yifan Wang[1,2], Audrey S. Richard[6], Hervé Raoul[6], Jiaming Lan[3], Markus A. Seeger[9], Yao Cong[1], Barry Rockx[6,7], Gary Wong[3,10✉], Yuhai Bi[2,4✉], Dimitri Lavillette[3,11✉] & Dianfan Li[1✉]

SARS-CoV-2, the causative agent of COVID-19[1], features a receptor-binding domain (RBD) for binding to the host cell ACE2 protein[1-6]. Neutralizing antibodies that block RBD-ACE2 interaction are candidates for the development of targeted therapeutics[7-17]. Llama-derived single-domain antibodies (nanobodies, ~15 kDa) offer advantages in bioavailability, amenability, and production and storage owing to their small sizes and high stability. Here, we report the rapid selection of 99 synthetic nanobodies (sybodies) against RBD by in vitro selection using three libraries. The best sybody, MR3 binds to RBD with high affinity ($K_D =$ 1.0 nM) and displays high neutralization activity against SARS-CoV-2 pseudoviruses ($IC_{50} =$ 0.42 μg mL$^{-1}$). Structural, biochemical, and biological characterization suggests a common neutralizing mechanism, in which the RBD-ACE2 interaction is competitively inhibited by sybodies. Various forms of sybodies with improved potency have been generated by structure-based design, biparatopic construction, and divalent engineering. Two divalent forms of MR3 protect hamsters from clinical signs after live virus challenge and a single dose of the Fc-fusion construct of MR3 reduces viral RNA load by 6 Log$_{10}$. Our results pave the way for the development of therapeutic nanobodies against COVID-19 and present a strategy for rapid development of targeted medical interventions during an outbreak.

[1] State Key Laboratory of Molecular Biology, CAS Center for Excellence in Molecular Cell Science, Shanghai Institute of Biochemistry and Cell Biology, Chinese Academy of Sciences (CAS), Shanghai, China. [2] University of CAS, Beijing, China. [3] CAS Key Laboratory of Molecular Virology & Immunology, Institut Pasteur of Shanghai CAS, Shanghai, China. [4] CAS Key Laboratory of Pathogenic Microbiology and Immunology, Institute of Microbiology, Center for Influenza Research and Early-warning (CASCIRE), CAS-TWAS Center of Excellence for Emerging Infectious Diseases (CEEID), CAS, Beijing, China. [5] Erasmus Laboratory Animal Science Center, Erasmus University Medical Center, Rotterdam, Netherlands. [6] European Research Infrastructure on Highly Pathogenic Agents (ERINHA-AISBL), Paris, France. [7] Department of Viroscience, Erasmus University Medical Center, Rotterdam, Netherlands. [8] National Facility for Protein Science in Shanghai, Shanghai Advanced Research Institute (Zhangjiang Laboratory), CAS, Shanghai, China. [9] Institute of Medical Microbiology, University of Zurich, Zurich, Switzerland. [10] Département de microbiologie-infectiologie et d'immunologie, Université Laval, Québec, QC, Canada. [11] Pasteurien College, Soochow University, Jiangsu, China. [12] These authors contributed equally: Tingting Li, Hongmin Cai, Hebang Yao, Bingjie Zhou, Ning Zhang. ✉email: garyckwong@ips.ac.cn; beeyh@im.ac.cn; dlaville@ips.ac.cn; dianfan.li@sibcb.ac.cn

The coronavirus disease that emerged in early December 2019 (coronavirus disease 2019 (COVID-19))[1] poses a global health and economic crisis[18]. The causative agent, severe acute respiratory syndrome coronavirus 2 (SARS-CoV-2), uses its Spike protein (S) to recognize receptors on host cells, an initial step for viral infection[1,2,19,20]. Key to this virus–host interaction is the binding between the S receptor-binding domain (RBD) and the host receptor angiotensin-converting enzyme 2 (ACE2)[3–6]. Therefore, the RBD has been a primary target for neutralizing antibodies that prevent ACE2 binding by either direct competition or steric hindrance[7–12,21].

Llama-derived nanobodies are generally more heat stable, easier and less expensive for production, and more amenable to protein engineering compared to conventional antibodies[22]. As single-chain antibodies, nanobody libraries are less complex to construct and screen, enabling in vitro selection of high-affinity binders in a relatively short time, typically 2–4 weeks[15,23–27]. Recently, several nanobody therapeutics, including the caplacizumab approved by the US Food and Drug Administration, have been developed for a variety of immune diseases[28]. Recent weeks have witnessed the generation of nanobodies that neutralize SARS-CoV-2 from several independent groups[13–17]. However, the in vivo efficacy of such nanobodies remains to be investigated.

Here we report the rapid selection of synthetic antibodies (sybodies)[24] in vitro using RBD as the bait. About half of the 99 sybodies neutralize SARS-CoV-2. Through structural and biophysical studies, we demonstrate that the sybodies SR4, MR17, and MR3 neutralize SARS-CoV-2 by blocking the RBD–ACE2 interaction. We improve the potency of MR17 and MR3 by structure-based mutagenesis or homo- and hetero-fusion. Finally, we show that prophylactic administration of nanobodies reduces viral loads and protects against pathological lung damages in hamsters. Our results form a preliminary basis for the development of nanobody therapeutics for COVID-19.

## Results and discussion

**Generation of high-affinity neutralizing sybodies against SARS-CoV-2.** SARS-CoV-2 S-RBD binders were selected by performing one round of ribosome display using three high-diversity libraries (Concave, Loop, and Convex)[24,25] and three rounds of phage display using the RBD as the bait under increasingly stringent conditions. Subsequent enzyme-linked immunosorbent assay (ELISA; Supplementary Fig. 1) identified 80, 77, and 90 positive clones, corresponding to 62, 19, and 18 unique binders from the Concave, Loop, and Convex library, respectively (Supplementary Data 1). None of the sybodies was identical to those obtained from the same libraries in two parallel studies[15,16], highlighting the high diversity of the libraries.

Eighty sybodies were further screened by a convenient fluorescence-detector size exclusion chromatography (FSEC) assay using the crude extract from sybody-expressing clones. This identified 28 (36%) sybodies, including 9 Concave (21%), 9 Loop (50%), and 10 Convex (56%) binders that caused earlier retention of the fluorescein-labeled RBD (Supplementary Fig. 2a and Supplementary Data 1).

All 99 sybodies were screened for neutralization activity against retroviral pseudotypes harboring the SARS-CoV-2 S protein. Using 50% neutralization at 1 μM concentration as a cut-off, 20 Concave (32%), 13 Loop (68%), and 10 Convex (56%) sybodies were identified as positive (Supplementary Fig. 3a). The high positive rates suggest high efficiency of the in vitro selection platform. Of note, none of the sybodies showed noticeable neutralization activities for the closely related SARS-CoV pseudovirus (Supplementary Fig. 3b). This is partly because antibodies recognizing three-dimensional (3D) epitopes are sensitive not only to epitope mutations but also to allosteric mutations that affect the conformational precision of the epitope. In line with this, cross-reactive antibodies against both SARS-CoVs reported so far[29–32] all exhibit much weaker binding toward the unintended CoV S-RBD, despite that they target conserved regions. Using strategies to block mutable regions during selection may help to develop cross-reactive sybodies.

Six FSEC-positive neutralizing sybodies, namely, SR4 (1), MR3 (31), MR4 (9), MR17 (1), LR1 (31), and LR5 (19) (S, M, L refers to Concave, Loop, and Convex sybodies, respectively; brackets indicate ELISA redundancy), were characterized in more detail as follows. They could be purified from *Escherichia coli* with a high yield (Fig. 1a). All six sybodies form a complex with the RBD (Supplementary Fig. 2b) with relatively high affinity (Fig. 1a and Supplementary Fig. 4) with $K_D$ ranging from 83.7 nM (MR17) to 1.0 nM (MR3). Consistent with its highest affinity, MR3 showed the slowest off-rate ($2.3 \times 10^{-4}$ s$^{-1}$). Using neutralization assays, we determined IC$_{50}$ of the six sybodies (Fig. 1b). MR3 was the most potent (IC$_{50}$ of 0.42 μg mL$^{-1}$), indicating a largely consistent trend between neutralization potency and binding kinetics (affinity and off-rate).

**Molecular mechanism for neutralization.** To gain mechanistic insights into neutralization, we performed crystallization trials for several RBD–sybody complexes and obtained crystals for four. Crystals of SR4- and MR17-RBD diffracted to 2.15 and 2.77 Å resolution, respectively, and allowed structure determination (Table 1). Crystals for MR3- and MR4-RBD did not diffract beyond 8.0 Å despite our optimization efforts.

The RBD structure resembles a short backrest high chair and SR4 binds to both the "seat" and "backrest" (Fig. 1c) with a surface area[33] of 727.4 Å$^2$ with modest electrostatic complementarity (Supplementary Fig. 5a). Of note, SR4 binds to RBD sideways, as intended by the design of the Concave sybody library[24]. All three complementarity determining regions (CDRs) contributed to the binding through hydrophobic interactions and H-bonding that involves both side chains and main chains (Fig. 1d). In addition, Tyr37, a framework residue, also participated in binding by forming an H-bond with the RBD Gly447 backbone.

MR17 also binds to the RBD at the "seat" and "backrest" regions but approaches the RBD at an almost perfect opposite direction of SR4 (Fig. 1c, e), indicating divergent binding mode for these sybodies. The binding of MR17 to the RBD occurred on an 853.9 Å$^2$ surface area with noticeable electrostatic complementarity (Supplementary Fig. 5b). Interestingly, this surface was largely shared with the SR4 binding surface (Fig. 1f). The interactions between MR17 and the RBD were mainly mediated by H-bonding. Apart from the three CDRs, two framework residues, Lys65 and Tyr60, interacted with the same RBD residue Glu484, via a salt bridge with its side chain and an H-bond with its main chain (Fig. 1g).

Structure alignment of SR4-, MR17-, and ACE2-RBD[3] showed that both sybodies engage with RBD at the receptor-binding motif (RBM) (Fig. 2a, b). Consistent with the structural observation, both SR4 and MR17 inhibited the binding of ACE2 to RBD, as revealed by biolayer interferometry (BLI) assays (Fig. 2c, d).

Aligning the SR4- and MR17-RBD structure with the "up"-RBD[2] in the "open"-S revealed no apparent hindrance between the sybody and S (Fig. 2e, f). In addition, owing to their minute sizes, SR4 and MR17 could be docked into the "close"-S with minor clashes against the Asn343-linked glycans from the anticlockwise neighboring RBD (Supplementary Fig. 6). Such clashes may be partly responsible for the modest neutralization activity of MR17 as its neutralizing activity increased by 17-fold

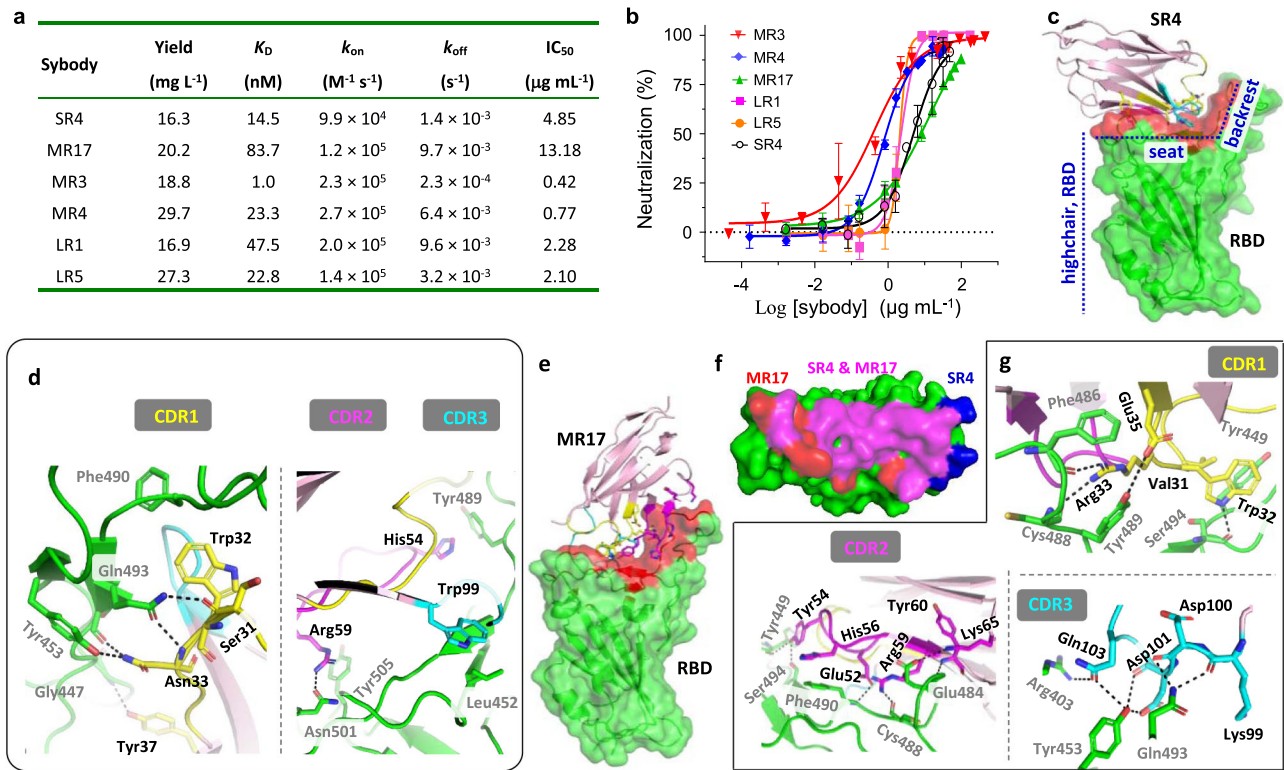

**Fig. 1 Biochemical and structural characterization of neutralizing synthetic nanobodies (sybodies). a** Summary of the characterization. Yield refers to purification from 1 L of culture. **b** Neutralization assay. SARS-CoV-2 pseudoviruses were preincubated with different concentrations of sybodies before infection of VeroE6-hACE2 cells. The rate of infection was measured by FACS. $IC_{50}$ was obtained by Sigmoidal fitting of the percentage of neutralization. Mean ± standard deviation are plotted ($n = 3$ or 4 independent experiments). Error bars are omitted where, in rare cases, available data points are less than three due to experimental design on concentration replicates. Color coding of the sybodies is as indicated. **c** The overall structure of SR4 (pink cartoon) in complex with RBD (green surface), which resembles a short backrest high chair. The binding surface is highlighted red. **d** SR4 CDR1 (yellow), CDR2 (magenta), and CDR3 (cyan) all contribute to the binding. Note that Tyr37 is a framework residue. **e** The overall structure of the MR17 (pink cartoon) in complex with RBD (green surface). The binding surface is highlighted red. **f** The overlap (magenta) between the SR4 (blue) and MR17 (red) interacting surfaces on RBD. **g** All three CDRs contribute to the binding with the receptor-binding domain (RBD) (green). Lys65 and Tyr60 are from the framework region. CDRs are color-coded as indicated in **d**. Dashed lines indicate H-bonding or salt bridges between atoms that are <4.0 Å apart. Black texts label sybody residues and gray texts label RBD residues. Source data for **b** are provided as a Source data file. CDR complementary determining region, FACS fluorescence-activated cell sorting, RBD receptor-binding domain.

upon elimination of the glycosylation by alanine mutation (Supplementary Fig. 6e). On the contrary, the N343A mutant was more resistant to SR4 than the wild type (Supplementary Fig. 6j), suggesting possible involvement of the glycans in SR4 binding in the context of the S trimer. As previously reported for other RBD-targeting antibodies[34], SR4 and MR17 showed tighter binding with the trimeric S (Fig. 2g, h) than with the monomeric RBD (Fig. 1a), likely reflecting the different oligomerization states between S and RBD.

The epitope of MR3 was probed using four different complementary methods. First, we performed cryo-electron microscopic (cryo-EM) single-particle analysis on the MR3-S complex, which had a dissociation constant of 1.7 nM (Supplementary Fig. 7a), and obtained a map at 6.25-Å resolution (Supplementary Figs. 7b and 8 and Supplementary Table 1). The major class features an S trimer with two RBDs assuming the "up" conformation (protomers A′ and B′) and the third one in the "down" conformation (protomer C′). Extra densities were observed on both "up"-RBDs, which we interpreted as MR3 engaging at the RBM region, and termed MR3-A′ and MR3-B′. Interestingly, the "down"-RBD appears to be occupied with a third MR3 molecule that clashes with the neighboring RBD and MR3 on protomer A′. This clashing may induce conformational heterogeneity, which could compromise the resolution of the

cryo-EM data. Second, we performed hydrogen–deuterium exchange mass spectrometry (HDX-MS) to probe RBD residues that are protected from HDX by MR3 binding. Mapping the protected residues onto the RBD structure displayed a surface that overlaps with RBM and extends further down at the side (Fig. 3a). Third, we conducted cross-competition assays using ACE2 and structurally characterized antibodies, including four monoclonal antibodies (mAbs; CB6[35], CV30[36], REGN10987, and REGN10933[12]), and three sybodies (SR4, MR17, and SR31[37]). MR3 competed with ACE2 and all the RBM antibodies (Fig. 3b–h) but not the non-RBM binder SR31 (Fig. 3i). Lastly, the epitope was assessed using three sets of alanine mutants (Fig. 3j, k and Supplementary Fig. 9). They include mutations of (1) RBM residues (Phe456, Gln493, Gln498, Tyr505), (2) RBM-peripheral residues (Arg346, Asn354, Arg403, Asp405, Asn450, Phe490), and (3) RBM-distal residues (Val367, Lys458, Glu465). Based on the BLI binding signal, the mutants were classified as severely impaired (0–25% of wild type), mildly affected (26–80%), and unaffected (>80%). Among the four RBM mutants, two (F456A, Y505A) displayed a severe reduction in BLI binding signal (Fig. 3j, k). The other two (Q493A, Q498A) showed a mild reduction in binding (Fig. 3k and Supplementary Fig. 9a). The RBM-peripheral group contains one severely impaired (R403A) and one mildly affected mutant (F490A) (Fig. 3j, k and

**Table 1 Data collection and refinement statistics.**

|  | SR4-RBD | MR17-RBD | MR17(K99Y)-RBD |
|---|---|---|---|
| *Data collection* |  |  |  |
| Space group | P $6_5$ 2 2 | P $3_2$ 2 1 | P $3_2$ 2 1 |
| Cell dimensions |  |  |  |
| $a, b, c$ (Å) | 65.55, 65.55, 344.53 | 73.69, 73.69, 158.58 | 74.19, 74.19, 158.40 |
| $\alpha, \beta, \gamma$ (°) | 90, 90, 120 | 90, 90, 120 | 90, 90, 120 |
| Wavelength (Å) | 0.97853 | 0. 97853 | 0.97853 |
| Resolution (Å) | 47.40–2.15 (2.23–2.15)[a] | 49.71–2.77 (2.89–2.77) | 49.90–2.94 (3.12–2.94) |
| $R_{merge}$ | 0.161 (1.203) | 0.276 (2.222) | 0.218 (1.666) |
| $R_{pim}$ | 0.054 (0.395) | 0.062 (0.494) | 0.052 (0.385) |
| $I/\sigma I$ | 11.4 (2.0) | 11.1 (1.5) | 12.4 (1.9) |
| Completeness (%) | 99.8 (99.9) | 100 (99.9) | 99.9 (99.6) |
| Multiplicity | 9.5 (9.9) | 21.0 (20.8) | 18.4 (19.3) |
| $CC^{\star}$[b] | 0.999 (0.970) | 0.997 (0.927) | 0.998 (0.920) |
| *Refinement* |  |  |  |
| Resolution (Å) | 47.40–2.15 | 49.71–2.77 | 49.90–2.94 |
| No. of reflections | 25,148 | 13,256 | 11,264 |
| $R_{work}/R_{free}$ | 0.1836/ 0.2239 | 0.2029/ 0.2659 | 0.2149/0.2676 |
| No. of atoms | 2810 | 2536 | 2509 |
| Protein | 2510 | 2482 | 2465 |
| Ligands | 62 | 54 | 44 |
| Water | 238 | 0 | 0 |
| No. of residues | 322 | 315 | 312 |
| B-factors (Å²) | 35.13 | 73.28 | 79.38 |
| Protein | 34.16 | 72.27 | 78.53 |
| Ligand/ion | 55.11 | 119.79 | 126.90 |
| Water | 40.22 |  |  |
| R.m.s. deviations |  |  |  |
| Bond lengths (Å) | 0.007 | 0.011 | 0.004 |
| Bond angles (°) | 0.850 | 1.10 | 0.68 |
| Ramachandran |  |  |  |
| Favored (%) | 98.06 | 96.12 | 96.08 |
| Allowed (%) | 1.94 | 3.88 | 3.59 |
| Outlier (%) | 0 | 0 | 0.33 |
| PDB ID | 7C8V | 7C8W | 7CAN |

[a]Highest resolution shell is shown in parenthesis.
[b]$CC^{\star} = \sqrt{\frac{2CC_{1/2}}{1+CC_{1/2}}}$.

of GS linkers ranging from 13 to 34 amino acids (Supplementary Data 1). Interestingly, the linker length had little effect on neutralization activity and these biparatopic LR5-MR3 sybodies were more potent than either sybodies alone (Fig. 1a) with an $IC_{50}$ of 0.11 μg mL$^{-1}$ (Fig. 4c). LR5-MR3 may be more tolerant to escape mutants[38–41] owing to its ability to recognize two distinct epitopes.

For Fc fusion, both MR3 and MR17 were separately attached to the dimeric human IgG Fc. This decreased $IC_{50}$ by 10-fold for Fc-MR3 (42 ng mL$^{-1}$) and 27-fold for Fc-MR17 (0.46 μg mL$^{-1}$), respectively (Fig. 4d, e). Consistently, the Fc fusion increased the apparent binding affinity for both sybodies, with a $K_D$ of 0.22 nM for Fc-MR3 and <1 pM for Fc-MR17 (Supplementary Fig. 4g, h). Note, however, Fc-MR17 did not gain as much neutralization potency as for the apparent binding affinity.

For tandem fusion, MR3 and a rationally designed MR17 mutant (MR17m, Supplementary Fig. 10) that showed comparable $IC_{50}$ with MR3 by a single mutation K99Y (0.49 μg mL$^{-1}$, Supplementary Fig. 10g) were individually linked together via GS linkers with variable length ranging from 13 to 34 amino acids (Supplementary Data 1). The optimal construct for MR17m-MR17m had the shortest linker (13 GS) (Fig. 4d, e). By contrast, optimal neutralization activity was observed with the longest linker (34 GS) for MR3-MR3 (Fig. 4d, e). Again, MR3-MR3 was superior compared to MR17m-MR17m, showing a 2-fold higher neutralization activity with an $IC_{50}$ of 10 ng mL$^{-1}$ (Fig. 4e). Compared to the monovalent MR3 ($IC_{50}$ of 0.42 μg mL$^{-1}$), the divalent engineering increased the potency by over 40-fold. Notably, MR3-MR3 showed similar activity to inhibit pseudotypes harboring the original SARS-CoV-2 S or the current dominant and more infectious mutant D614G S[42] (Fig. 4d).

**In vivo protection for SARS-CoV-2 by divalent MR3.** To investigate the in vivo protection of the sybodies against SARS-CoV-2 live virus, we first conducted a preliminary experiment using mice. The most potent divalent sybody (MR3-MR3) was chosen. Nanobodies have very short serum half-lives of several minutes due to their minute size[43]. To circumvent this, we fused MR3-MR3 to the N-terminus of an albumin-binding domain (ABD)[44], which has been known to extend the circulating half-life of its fusion partners by increasing size and preventing intracellular degradation[28].

MR3-MR3-ABD bound to the human albumin (Supplementary Fig. 11a) while retaining its ability to bind RBD (Supplementary Fig. 11b) and neutralize SARS-CoV-2 pseudotypes harboring either past (614D) and current (614G)[42] SARS-CoV-2 S (Fig. 5a). In addition, when assayed with authentic SARS-CoV-2, MR3-MR3-ABD showed ~60% neutralization at 20 ng mL$^{-1}$ (Fig. 5b). As designed, a serum virus neutralization assay showed that the addition of the ABD to the divalent MR3 (MR3-MR3-ABD) extended its in vivo stability in mice, displaying neutralization activity up to 24 h post-injection contrary to MR3 or MR3-MR3 (Supplementary Fig. 11c). The body weight measures and the microscopic histopathology analysis did not reveal any toxicity for the nanobodies for 6 days (Supplementary Fig. 11d, e).

To test the in vivo antiviral efficacy of MR3-MR3-ABD, C57BL/6J female mice, aged 6–8 weeks, were first sensitized to SARS-CoV-2 infection using an adenovirus expressing the human ACE2 receptor[45] at 5 days before challenge. Mice were infected via the intranasal route with $5 \times 10^5$ median tissue culture infectious dose ($TCID_{50}$) of SARS-CoV-2 and then administered a single dose of 25 mg kg$^{-1}$ MR3-MR3-ABD via the intraperitoneal (IP) route at 12 h after virus challenge. A control group was given phosphate-buffered saline (PBS) as a mock

Supplementary Fig. 9a). The rest of this group and all three in the RBM-distal group displayed a similar binding profile as the wild type (Fig. 3k and Supplementary Fig. 9b). It should be noted that Lys458, although being identified as a participating residue in the HDX-MS experiment (Fig. 3a), is unlikely a major contributor to the MR3 binding based on the mutagenesis results (Fig. 3k). Overall, these results provided mutually corroborative evidence that MR3, like SR4 and MR17, targets RBD at the RBM surface and neutralizes SARS-CoV-2 by competitively blocking the ACE2–RBD interaction.

**Sybody engineering increases affinity and neutralizing activity.** Increasing valency is a common technique to enhance potency for nanobodies[17,28]. To this end, we engineered three types of divalent sybodies, including the biparatopic fusion of two different sybodies, and the Fc-fusion and tandem fusion of the same sybody.

For biparatopic fusion, we first identified two sybodies, namely, LR1 and LR5 (Fig. 4a, b), that could bind RBD in addition to MR3 using the BLI assay. As LR5 showed higher affinity (Fig. 1a and Supplementary Fig. 4e, f) than LR1, we fused this non-competing sybody to the N-terminal of MR3 with various lengths

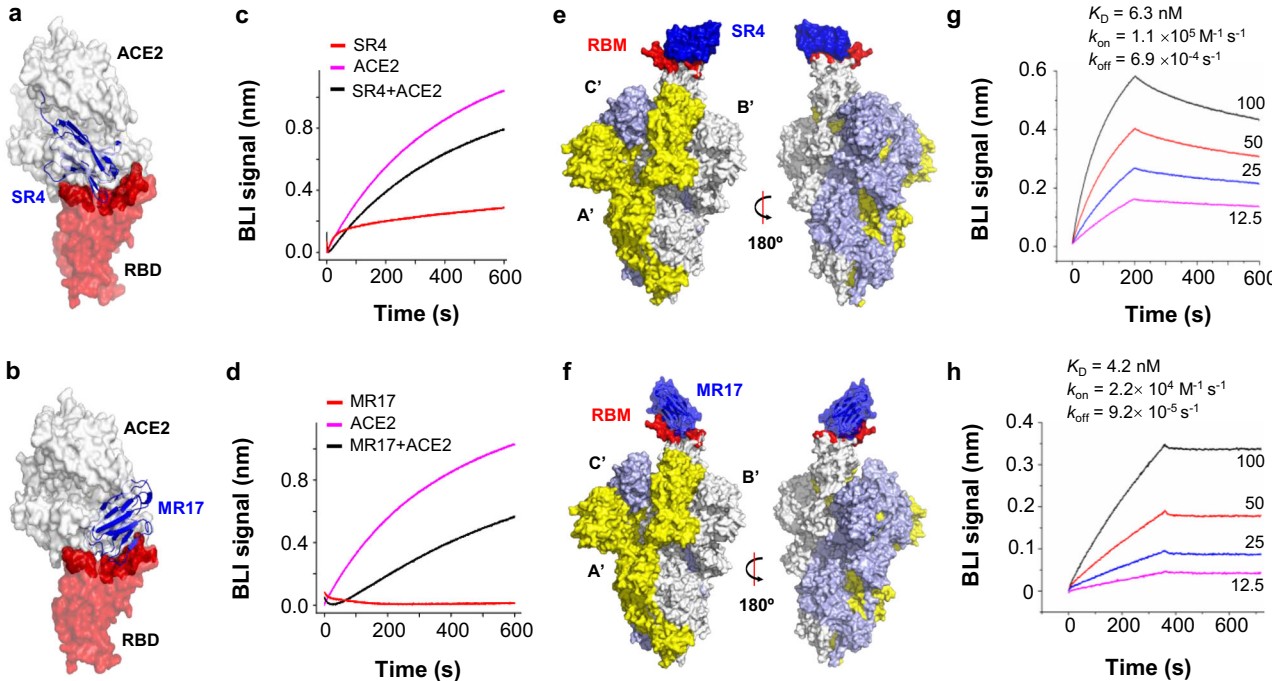

**Fig. 2 Molecular basis for neutralization. a, b** Alignment of the SR4-RBD (**a**) or MR17-RBD (**b**) to the ACE2-RBD structure (PDB ID 6M0J)[3] reveals that SR4/MR17 (blue) binds RBD (red) at the motif (dark red) where ACE2 (white) also binds. **c, d** SR4 (**c**) and MR17 (**d**) compete with ACE2 for RBD binding. A sensor coated with streptavidin was saturated with 2 µg mL[−1] of biotinylated RBD. The sensor was then soaked in 200 nM of the indicated sybody before further soaked in sybody-containing buffer with (black) or without (red) 25 nM of ACE2 for BLI signal recording. As a control, the ACE2–RBD interaction was monitored in the absence of sybodies (magenta). **e, f** Alignment of the SR4-RBD (**e**) and MR17-RBD (**f**) to the "up" conformation of the RBD from the cryo-EM structure of the trimer S (PDB ID 6VYB)[2]. The three subunits are colored yellow (A'), white (B'), and light blue (C'). Synthetic nanobodies are colored deep blue. RBM (red) marks the ACE2-binding motif. **g, h** Binding kinetics of SR4 (**g**) and MR17 (**h**) to S. BLI assay was performed with sybodies immobilized and S as analyte at the indicated concentrations (nM). Source data for **c, d, g, h** are provided as a Source data file. ACE2 angiotensin-converting enzyme 2, BLI biolayer interferometry, RBD receptor-binding domain, RBM receptor-binding motif.

treatment. Virus titer analysis (Supplementary Fig. 12a, b) and histopathology examinations (Supplementary Fig. 12c–e) suggest a tendency of protection by MR3-MR3-ABD, although statistical conclusions could not be drawn because the differences between the two groups were significant for the $TCID_{50}$ results (Supplementary Fig. 12b) but not for the quantitative reverse transcription polymerase chain reaction (qRT-PCR) data (Supplementary Fig. 12a).

To further investigate the in vivo efficacy of MR3, we repeated the animal experiment using hamsters, which is a better model than mice for COVID-19 because hamsters develop severe symptoms upon infection[46]. In addition to MR3-MR3-ABD, Fc-MR3, which showed ~90% neutralization for authentic SARS-CoV-2 at 20 ng mL[−1] (Fig. 5b), was also used, along with non-neutralizing sybodies (Sb92-Sb44-ABD and Fc-Sb2 for the ABD- and Fc-fusion, respectively) as controls. Hamsters were administrated via the IP route with 2.5 mg of divalent sybodies 6 h before infection with $10^5$ $TCID_{50}$ of SARS-CoV-2 (strain BetaCoV/Munich/BavPat1/2020).

Compared to non-infected hamsters, the virus challenge caused severe weight loss (~20%) by 4 days post-infection (dpi). Prophylactic IP injection of MR3-MR3-ABD reduced the weight loss by ~50%. Remarkably, despite initial weight loss in the first 2 days, the Fc-MR3 group regained weight to as much as the non-infected group. As controls, neither Sb44-Sb92-ABD nor Fc-Sb2 showed any protection (Fig. 5c). The viral RNA load in the lung was reduced by approximately sevenfold by MR3-MR3-ABD, compared to the Sb92-Sb44-ABD group. Consistent with the weight loss results, the injection of Fc-MR3 reduced the RNA load by a dramatic 6 $Log_{10}$, falling to the detection limit (Fig. 5d).

Infectious virus was not detectable in the lungs of hamsters treated with Fc-MR3 or in three out of five hamsters treated with MR3-MR3-ABD (Fig. 5e). Despite the strong inhibition of the virus replication in the lung, the sybody treatment did not reduce viral RNA load in nasal turbinates, although infectious titers were below the limit of detection (Fig. 5f, g). Finally, histopathologic analysis confirmed that MR3-MR3-ABD offers modest protection for the hamsters from lung damage, but Fc-MR3 offers almost full protection from SARS-CoV-2 infection (Fig. 5h, i).

There is some variability in viral titers and pathology for the hamster experiments. Specifically, several data points unexpectedly fall below the detection limits (Fig. 5e, g), probably causing the discrepancies between the virus load (Fig. 5d, f) and infectious titer. It should be noted that the detection of SARS-CoV-2 is most sensitive when using qRT-PCR. However, in the context of potential for transmission, we also performed virus titrations on homogenates of hamster lung and nasal turbinates, to detect the presence of infectious viruses. Unfortunately, this method is less sensitive (limit of detection 1000 $TCID_{50}$ per milliliter). In addition, variation in infectious titers is increased when normalizing to grams of tissue. Finally, because no fully inbred strain of hamsters is available, some biological variations may be expected. Taking into account inherent limitations of quantitative recovery of infectious virus from tissue homogenates, these represent viral replication in target tissues. Statistically significant conclusions for the viral titer in hamsters were only based on quantitative measures of viral RNA.

Interestingly, while the MR3-MR3-ABD was similarly potent with Fc-MR3 in vitro, it was less efficient than Fc-MR3 when tested in hamsters. This may be related to their difference in

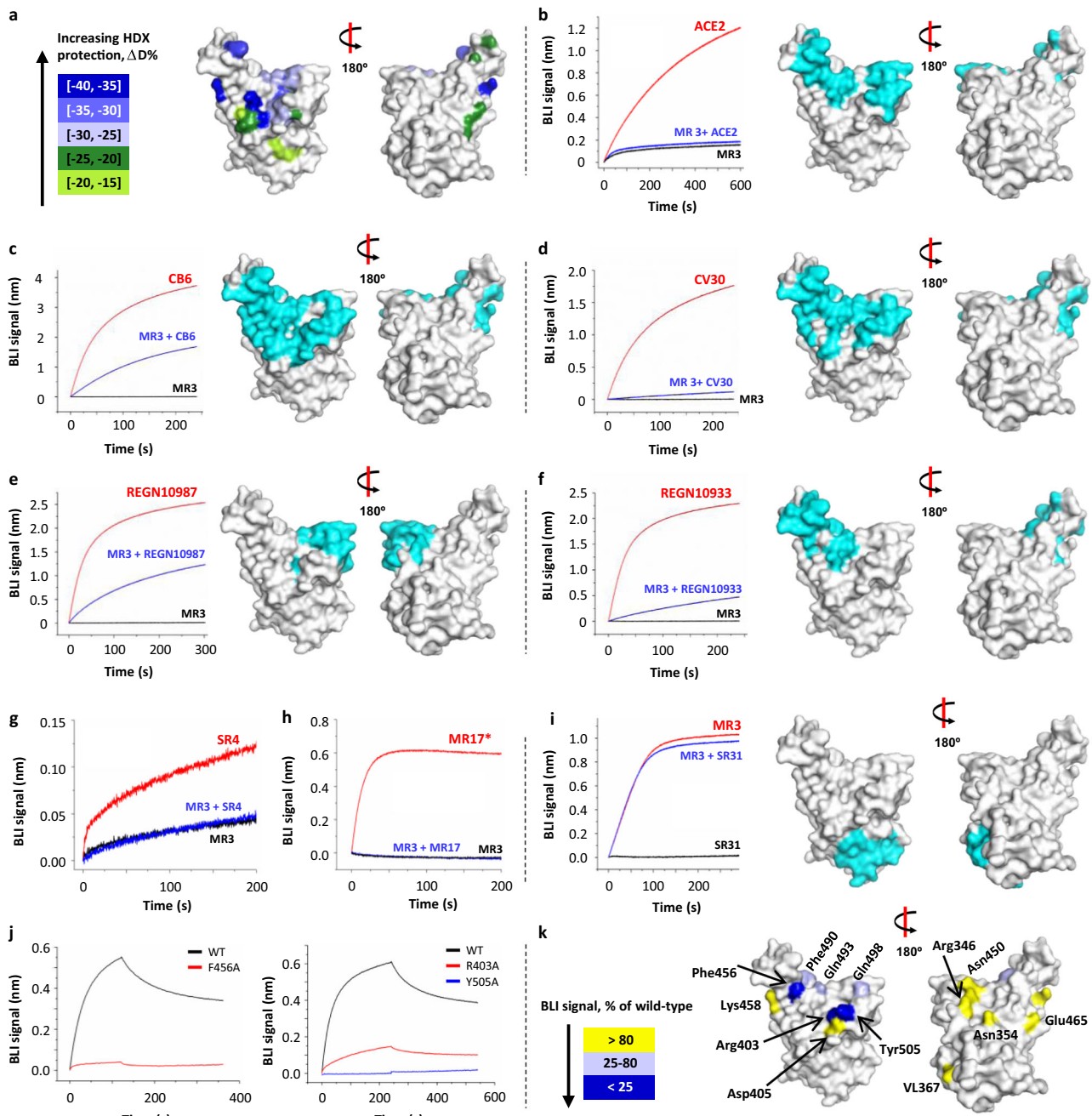

**Fig. 3 HDX-MS, binding, and mutagenesis studies suggest an overlap between the MR3 epitope and RBM. a** Epitope mapping using HDX-MS. RBD residues are color-coded based on the extent of the solvent exchange as an indication of protection by MR3-binding. **b** MR3 competes with ACE2 for RBD binding. The RBM surface is highlighted in cyan. **c–h** MR3 competes with RBM-targeting antibodies for RBD binding. They include monoclonal antibodies CB6[35] (**c**), CV30[36] (**d**), REGN10987[12] (**e**), REGN10933[12] (**f**), and the two sybodies SR4 (**g**) and MR17 (**h**) characterized in this study. **i** The MR3-RBD binding is compatible with SR31, which targets a non-RBM surface[37]. In **c–h**, the epitopes of the antibodies are highlighted in cyan except for SR4 and MR17, which are shown in Fig. 1c, e. In **b–h**, binding assays were performed using RBD immobilized on a sensor. BLI profiles were recorded with (black and blue) or without (red) pre-saturation of MR3 using the indicated antibodies as analytes. The assays in **i** were performed as in **b–h** except that SR31 was used for pre-saturation. **j** Three RBD mutations (red and blue) with severely impaired MR3 binding compared with the wild type (WT, black). The data for the rest of the 10 mutants are in Supplementary Fig. 9. **k** Mapping the mutagenesis data to the RBD structure. Residues are colored based on the sensitivity to alanine mutation; mutants showing <80% binding signal compared to the wild type are considered to be involved in binding, assuming no allosteric effects. Source data for **b–j** are provided as a Source data file. ACE2 angiotensin-converting enzyme 2, BLI biolayer interferometry, HDX-MS hydrogen–deuterium exchange mass spectrometry, RBD receptor-binding domain, RBM receptor-binding motif.

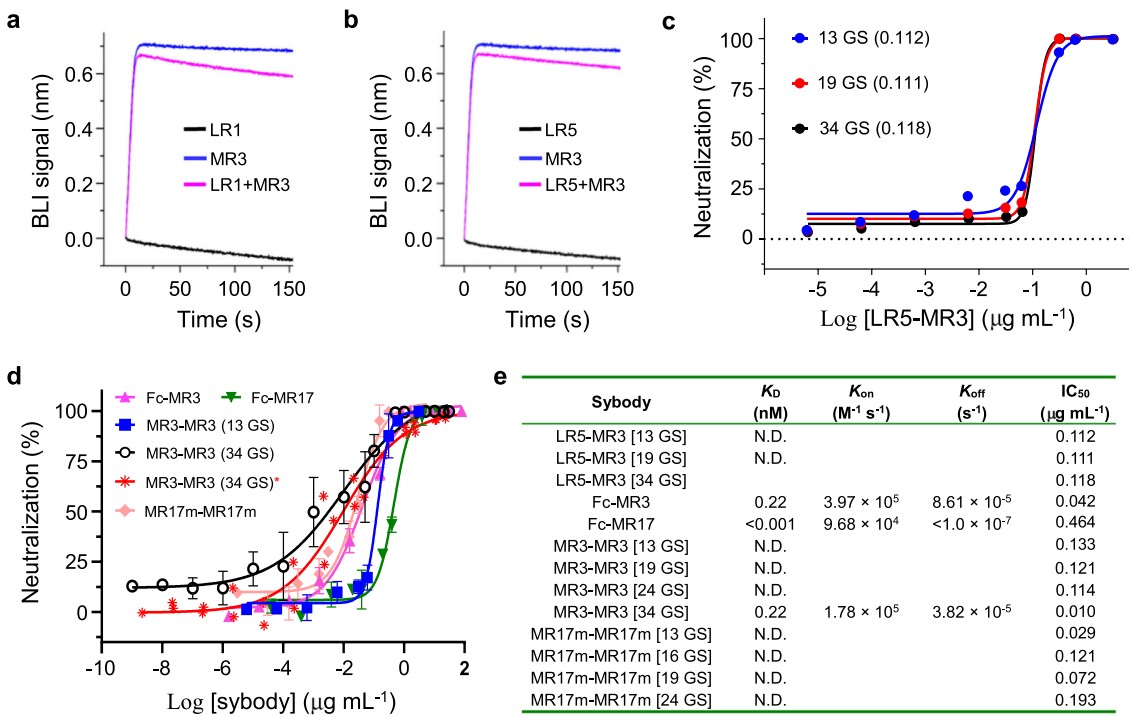

**Fig. 4 Divalent engineering increases affinity and neutralizing activity. a, b** Identification of two non-competing pairs, LR1/MR3 (**a**) and LR5/MR3 (**b**), for biparatopic constructs. For BLI assays, sensors coated with RBD were soaked in 200 nM of LR1 or LR5 before further soaked in LR1- or LR5-containing buffer with (magenta) or without (black) 100 nM of MR3. The MR3–RBD interaction profile was obtained in the absence of LR1 or LR5 (blue). **c** Neutralization assay of the biparatopic sybody LR5-MR3 with a Gly-Ser (GS) linker of 13 (blue), 19 (red), or 34 (black) amino acids as indicated. Brackets indicate IC50 values in μg mL$^{-1}$. Data are from a representative of two independent experiments. **d** Neutralization assays of divalent sybodies. The original SARS-CoV-2 was used for all assays except that the D614G mutant[42] was additionally tested for MR3-MR3 (red asterisk). Color-coding of the tested sybodies is as indicated. For MR3-MR3(34GS) against 614G, data show a representative of three independent experiments, which were performed using non-overlapping concentrations. For the rest of the samples, mean ± standard deviation are plotted ($n = 3$ independent experiments). Error bars are omitted where, in rare cases, available data points are less than three due to experimental design on concentration replicates. **e** Summary of binding kinetics and neutralizing activities of the divalent sybodies. Source data for **a–d** are provided as a Source data file. BLI biolayer interferometry, N.D. not determined.

half-life. As shown in Supplementary Fig. 11c, Fc-MR3 had a much longer half-life than MR3-MR3-ABD, although we note that the stability was tested in mice (Supplementary Fig. 11c) while the protection was compared in hamsters (Fig. 5c–i). The possible difference of tissue distribution of the sybody forms, and pharmacokinetic stability modulated by the binding affinity to Fc receptors and the binding affinity to albumin from different species, may need to be investigated to explain their differences in in vivo efficiency.

Owing to their high stability, nanobodies can survive nebulization. Of relevance to COVID-19, an inhaled nanobody drug (ALX-0171) has gone into clinical trials for the treatment of the respiratory syncytial virus[28]. Because of the high stable framework as originally designed[26], the potential of the neutralizing sybodies for the development of inhaled therapy warrants future investigations.

In summary, the in vitro platform was efficient in generating neutralizing sybodies (the selection process took 2 weeks). Structural and biochemical studies suggest an antagonistic mechanism to block the ACE2–RBD interaction. Protein engineering has yielded various forms of sybodies with higher affinity, neutralization activity, and in vivo stability (in mice). Using the most potent construct, we have demonstrated that nanobodies can provide prophylactic protection of hamsters from SARS-CoV-2 infection. Our results should encourage the development of nanobody therapeutics to fight COVID-19 or future viral outbreaks.

## Methods

**Protein expression and purification—SARS-CoV-2 S-RBD for sybody selection.** The construct for the RBD with an Avi-tag for biotinylation was made by fusing DNA, from 5'- to 3'-end, of the encoding sequence for the honey bee melittin signal peptide (KFLVNVALVFMVVYISYIYAA), a Gly-Ser linker, residues of 330–531 of the SARS-CoV-2 spike protein (Uniprot ID P0DTC2), a Gly-Thr linker, the 3C protease site (LEVLFQGP), a Gly-Ser linker, the Avi tag (GLNDI-FEAQKIEWHE), a Ser-Gly linker, and a deca-His tag into a pFastBac-backbone vector by Gibson assembly[47] using primers listed under "Gibson assembly for RBD" in Supplementary Data 2. Baculoviruses were generated in *Spodoptera frugiperda sf*9 cells using standard Bac-to-Bac protocols and expression was achieved by infecting *Trichoplusia ni* High Five suspension cells at $2 \times 10^6$ cells per milliliter for 48–60 h at 27 °C in flasks. The medium from 1 L of culture was filtered through a 0.22-μm membrane and incubated with 3.0 mL of Ni-Sepharose Excel (Cat. 17-3712-03, GE Healthcare) in the presence of 20 mM of imidazole for 2–3 h at 4 °C with mild agitation. The beads were washed with 10 column volume (CV) of 20 mM imidazole in Buffer A (150 mM NaCl, 20 mM Tris HCl pH 8.0). The RBD was eluted using 300 mM of imidazole in Buffer A. For biotinylation[24,25], the purified RBD with the Avi-tag intact (0.8 mg mL$^{-1}$) was incubated with 5 mM ATP, 10 mM magnesium acetate, 43.5 μM biotin, and 22 μg mL$^{-1}$ home-purified BirA in 3.2 mL volume and incubated at 4 °C for 16 h. Biotinylated RBD was concentrated using a 10-kDa cut-off membrane concentrator to ~3 mg mL$^{-1}$ before loaded onto a Superdex Increase 200 10/300 GL column for SEC. Fractions containing the RBD were pooled, ali-quoted, flash-frozen in liquid nitrogen, and stored at −80 °C before use.

For RBD mutants, plasmids containing desired mutations were generated by standard PCR-based site-directed mutagenesis methods using primers listed under "RBD mutations" in Supplementary Data 2. RBD mutants were expressed and purified the same way as the wild type except that the gel filtration step was replaced with desalting into PBS buffer on a Bio-Rad Econo-Pac 10DG column.

**Protein expression and purification—SARS-CoV-2 S-RBD for crystallization.** For protein crystallization, the RBD was purified as above. Both the Avi-tag and the His-tag were removed by 3C protease digestion as follows. The pooled elution from

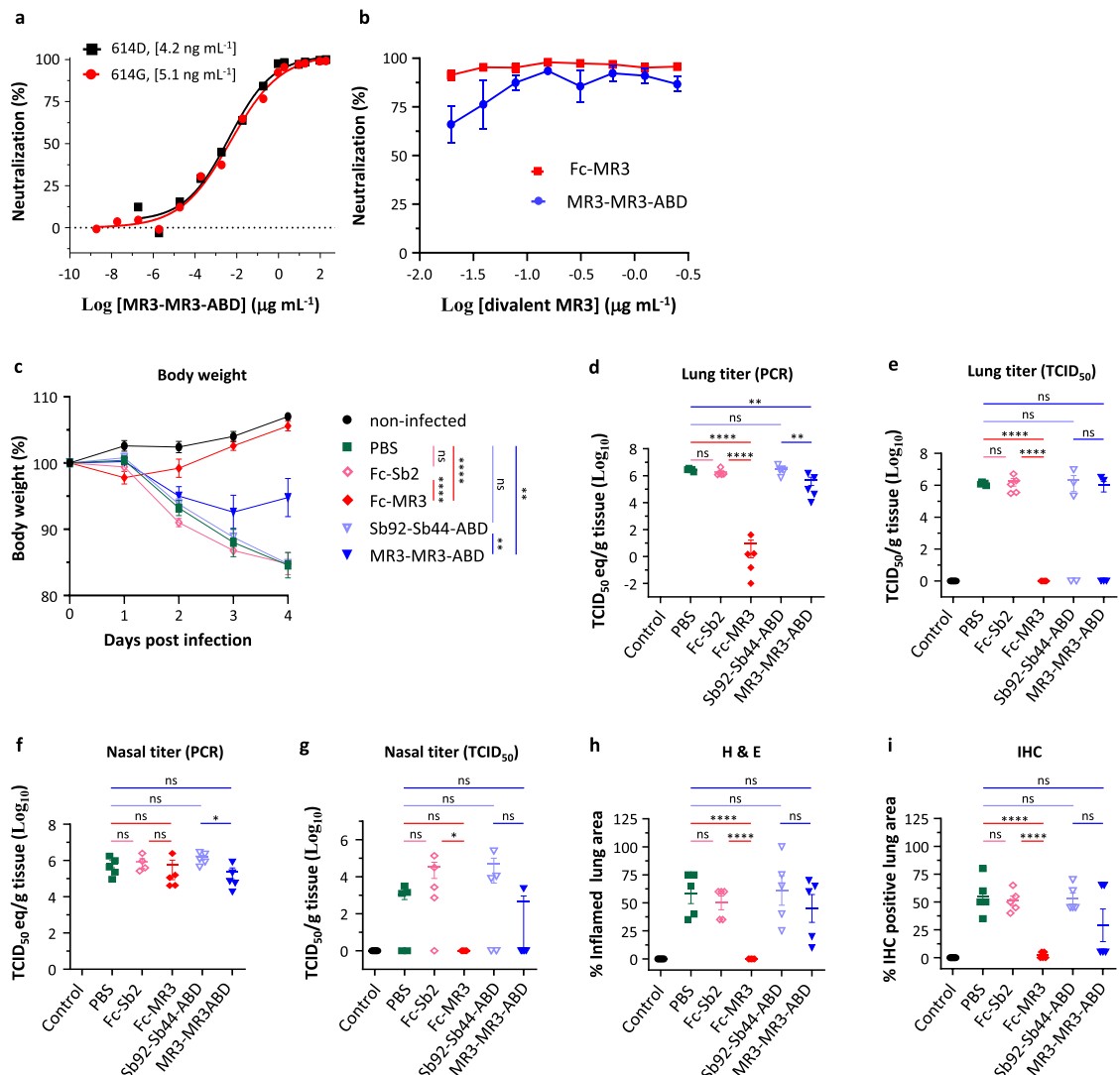

**Fig. 5 Potent divalent MR3 protects hamsters against weight loss and viral replication. a** Neutralization assay of MR3-MR3-ABD using pseudovirus bearing the wild-type Spike (614D, black square) or the D614G mutant Spike (614G, red circle). Data are from one representative experiment of three (614 G) or two (614D) independent experiments. Statistics were not performed for 614G because the three experiments were performed with different sybody concentrations. **b** Neutralization of authentic SARS-CoV-2 by Fc-MR3 (red square) and MR3-MR3-ABD (blue circle) measured using a plaque-reduction assay. Mean ± standard deviation are plotted ($n = 3$ independent experiments). **c** Body weights of hamsters treated with antibodies (color-coded as indicated) were measured at the indicated days after inoculation with SARS-CoV-2. Statistics were performed using two-way ANOVA followed by Sidak's multiple comparisons test. $**p < 0.005$; $****p < 0.001$. Timepoint starting to show significance: PBS vs Fc-MR3 and Fc-Sb2 vs Fc-MR3, 2 dpi; PBS vs MR3-MR3-ABD and Sb92-Sb44-ABD vs MR3-MR3-ABD, 4 dpi. SARS-CoV-2 viral RNA (**d**, **f**) or infectious virus (**e**, **g**) was detected in the lung (**d**, **e**) and the nasal turbinates (**f**, **g**). Percentage of inflamed lung tissue (**h**) and percentage of lung tissue expressing SARS-CoV-2 antigen (**i**) estimated by microscopic examination in different groups of hamsters at 4 days after SARS-CoV-2 inoculation. In **c**–**i**, the mean percentage of starting weight, the mean copy number, the mean infectious titer, or the mean pathology score is shown, error bars represent the standard error of the mean (SEM). $n = 5$ biologically independent samples. Statistical analyses were performed using one-way ANOVA followed by Tukey's multiple comparisons test on log-transformed data (**d**–**g**) or raw data (**h**, **i**). ns not significant; $*p < 0.05$; $**p < 0.01$; $****p < 0.0001$. Source data of **a**–**i** and exact $p$ values of **c**–**i** are provided as a Source data file.

Ni-Sepharose Excel column was desalted to remove imidazole using a desalting column (Cat. 732-2010, Bio-Rad) pre-equilibrated in Buffer A. The desalted RBD was mixed with home-purified His-tagged 3C protease at 1:100 molar ratio (3C protease:RBD) at 4 °C for 16 h. The mixture was then passed through a Ni-NTA column, which binds 3C protease, undigested RBD, and the cleaved His-tag. The flow-through fractions were collected and concentrated to 8–10 mg mL$^{-1}$. The protein was either used directly for crystallization or flash-frozen in liquid nitrogen and stored at −80 °C before use.

For crystallization, fresh RBD or thawed from −80 °C was mixed with desired sybodies at 1:1.5 molar ratio (RBD:sybody). After incubation on ice for 30 min, the mixture was clarified by centrifugation before SEC. Fractions containing the complex were pooled, concentrated to ~10–15 mg mL$^{-1}$ before crystallization trials.

**Protein expression and purification—SARS-CoV-2 Spike**. The plasmid for mammalian transient expression of S harbors the mammalian codon-optimized gene encoding residues Met1–Gln1208 of the SARS-CoV-2 S with mutations K986P, V987P, a GSAS linker substituting Arg682-Arg685 (the furin cleavage site), a C-terminal T4 fibritin trimerization motif (GYIPEAPRDGQAYVRKDGEWV LLSTFL), a TEV protease cleavage site, a FLAG tag, and a polyhistidine tag[48]. For expression, Expi293 cells were transfected with a mixture containing the plasmid and polyethylenimine (PEI). After 3.5 days of suspension culturing, the medium was removed from the cells by filtration through a 0.22-μm membrane and adjusted to have 200 mM NaCl, 20 mM imidazole, 4 mM MgCl$_2$, and 20 mM Tris-HCl pH 7.5. The filtrate was added with 3 mL of Ni-NTA beads and incubated at 4 °C for 2 h. The beads were loaded into a gravity column and washed with 50 CV of 20 mM imidazole, before eluted with 250 mM imidazole in 200 mM NaCl, and

20 mM Tris-HCl pH 7.5. The eluted fractions were pooled, concentrated using a 100-kDa cut-off membrane concentrator, and further purified using SEC. Fractions containing S were pooled, quantified using a $\varepsilon_{280}$ of 138,825 $M^{-1}$ $cm^{-1}$, concentrated to 2 mg $mL^{-1}$, and mixed with a 4-fold molar concentration of MR3 sybody for cryo-EM sample preparation.

**Protein expression and purification—sybodies in *E. coli*.** Sybodies were expressed with a C-terminally His-tag in *E. coli* MC1061 cells. Briefly, cells carrying sybody genes in the vector pSb-init[24,25] were grown in Terrific Broth (TB, 0.017 M $KH_2PO_4$ and 0.072 M $K_2HPO_4$, 1.2% (w/v) tryptone, 2.4% (w/v) yeast extract, 0.5% (v/v) glycerol) supplemented with 25 mg $L^{-1}$ chloramphenicol to $OD_{600}$ of 0.5 at 37 °C in a shaker-incubator at 220 rpm. The growth temperature was lowered to 22 °C and the cells were allowed to grow for another 1.5 h before induced with 0.02% (w/v) arabinose for 17 h. Cells were lysed by osmotic shock. Briefly, cells from 1 L of culture were resuspended in 20 mL of TES-high Buffer (0.5 M sucrose, 0.5 mM EDTA, and 0.2 M Tris-HCl pH 8.0) and incubated at 4 °C for 30 min. After this dehydration step, cells were abruptly rehydrated with 40 mL of ice-cold MilliQ $H_2O$ at 4 °C for 1 h. The periplasmic extract released by the osmotic shock was collected by centrifugation at 20,000 × g at 4 °C for 30 min. The supernatant was adjusted to contain 150 mM of NaCl, 2 mM of $MgCl_2$, and 20 mM of imidazole before added with Ni-NTA resin that had been pre-equilibrated with 20 mM of imidazole in Buffer A (150 mM NaCl and 20 mM Tris HCl pH 8.0). After batch-binding for 2 h, the beads were washed with 30 mM imidazole, before eluted with 300 mM imidazole in Buffer A. The eluted protein was either used directly or flash-frozen in liquid nitrogen and stored at −80 °C.

**Protein expression—sybody MR3-MR3-ABD in *Pichia pastoris*.** The encoding gene for MR3-MR3-ABD (Supplementary Data 1) was cloned into vector pPICZαC (Invitrogen) immediately in frame with the α-factor signal peptide. To express MR3-MR3-ABD in yeast, *P. pastoris* GS115 and SMD1168H were transformed with *Sac*I-linearized plasmid and selected with 0.1 and 0.5 mg $mL^{-1}$ zeocin on an YPDS agar plate (1% (w/v) yeast extract, 2% (w/v) peptone, 2% (w/v) glucose, 0.8 M sorbitol, 2% (w/v) agarose). Colonies (12 for each strain) were inoculated into 3 mL YPD liquid medium. Cells were cultured in a 30-°C incubator. After 24 h, cells were harvested, washed twice with methanol complex medium (BMMY), and suspended in BMMY medium at a final $OD_{600}$ of 4–5 for induction. Methanol was supplemented to the medium to 0.5% (v/v) every 24 h. After 3 days of expression, the medium was collected by centrifugation and the secreted protein was used for sodium dodecyl sulphate–polyacrylamide gel electrophoresis analysis.

To quantify the expression level, the supernatant (10 μL) was loaded together with known amounts of MR3-MR3-ABD (purified from *E. coli*, 0.5, 1, 2, 3, and 4 μg) that had been pre-mixed with medium from the culture of untransformed GS115. The band intensity was semi-quantified by densitometry analysis using the Image Lab 5.2 software (Bio-Rad).

**Protein expression and purification—divalent sybodies in mammalian cells.** The encoding sequence of MR3 was cloned into a vector harboring the hinge and Fc regions of IgG2 (Supplementary Data 1, Uniprot P01859) for secretion in mammalian cells. Expi293 cells at a density of 2.3 million per milliliter were transfected with the plasmid (final concentration of 2 mg $L^{-1}$) using linear PEI (average molecular weight of 25 kDa, 4 mg $L^{-1}$). Sodium valproate was included at a final concentration of 2 mM. Cells were cultured in a flask for 65 h. The supernatant was collected by centrifugation and filtered through a 0.22-μm membrane. The filtrate from 2 L of culture was incubated with 3.2 mL rProtein A beads (Cat. SA012005, SmartLifesciences, China) for batch binding at 4 °C for 3 h. The beads were packed into a gravity column, washed with 20 CV of PBS buffer before eluted with 0.1 M glycine pH 3.0. The elution was quickly neutralized using 1 M Tris HCl pH 8.0. The buffer was then exchanged to PBS using a desalting column.

**Protein expression and purification—mAbs.** The plasmid harboring encoding genes for mAbs CB6 was a kind gift from Professor Bing Sun at the last author's institute. DNA encoding the variable region of mAbs CV30, REGN10987, and REGN10933 were synthesized and Gibson assemblied[47] into the vector pDEC. The plasmids (1.4 mg (light chain) and 0.6 mg (heavy chain) per liter of cells at ~2 × $10^6$ $mL^{-1}$ density) were transfected into Expi293 cells using PEI. Three days after transfection, the medium was harvested by centrifugation at 1500 × g for 15 min, followed by filtration through a 0.22-μm membrane. The filtrate containing secreted antibodies was incubated with Protein A beads for 2 h at 4 °C. The beads were washed with 20 CV of PBS before eluted with a buffer containing 0.1 M glycine-HCl pH 3.0. After elution, Tris-HCl at pH 8.0 was added to a final concentration of 0.1 M and NaCl to 0.15 M. The buffer was exchanged to PBS on a desalting column (Econo-Pac 10DG, Bio-Rad). The antibodies were quantified using absorbance at 280 nm and the theoretical extinction coefficient was calculated based on the content of aromatic residues.

**Sybody selection—ribosome display and phage display.** Sybody selection was performed using a combination of ribosome display and phage display[24,25]. In vitro translation of the "Concave," "Loop," and "Convex" library was performed according to the manufacturer's instruction (PURE*frex* 2.1 kit, Cat. PF213-0.25-EX,

Genefrontier, Chiba, Japan). A reaction mix containing 1.8 μL of nuclease-free water, 4 μL of solution I, 0.5 μL of solution II, 1 μL of solution III, 0.5 μL of 10 mM cysteine, 0.5 μL of 80 mM reduced glutathione, 0.5 μL of 60 mM oxidized glutathione, and 0.5 μL of 1.875 mg $mL^{-1}$ disulfide bond isomerase DsbC (DS supplement, Cat. PF005-0.5-EX, Genefrontier) was warmed at 37 °C. After 5 min, 0.7 μL of mRNA library, corresponding to $1.6 \times 10^{12}$ mRNA molecules, was added to the pre-warmed mix for in vitro translation at 37 °C for 30 min. The reaction was diluted with 100 μL ice-cold Panning Solution (150 mM NaCl, 50 mM magnesium acetate, 0.5% (w/v) bovine serum albumin (BSA), 0.1% (w/v) Tween 20, 0.5% (w/v) heparin, 1 μL RNaseIn, and 50 mM Tris-acetate pH 7.4) and cleared by centrifugation at 20,000 × g for 5 min at 4 °C. Biotinylated RBD was added to the supernatant and the mixture was incubated on ice for 20 min. Streptavidin beads (Dynabeads Myone Streptavidin T1) were added to pull down the complex consisting of nascent sybody that binds to RBD, the stalled ribosome with the mRNA encoding the binders, and biotinylated RBD. Selected mRNAs were purified and reverse-transcribed into single-chain DNA with the primer[24] RevTranscript (Supplementary Data 2) using reverse transcriptase (Cat. 200436, Agilent). The resulting cDNA library was purified using a DNA Purification Kit (Cat. A740609.25, Macherey-Nagel) and PCR-amplified using the primer pair Concave_Loop_FX_fwd and Concave_Loop_FX_rev for "Concave" and "Loop" libraries and the primer pair Convex_FX_fwd and Convex_FX_rev (Supplementary Data 2) for the "Convex" library. The product was gel-purified, digested with the Type IIS restriction enzyme *Bsp*QI, and ligated into the vector pDX_init[24,25] treated with the same enzyme. The ligation product was then transformed into *E. coli* SS320 competent cells by electroporation to generate libraries for phage display.

Three rounds of phage display were carried out. The first round was performed in a 96-well plate coated with 67 nM neutravidin (Cat. 31000, Thermo Fisher Scientific). Phage particles were incubated with 50 nM biotinylated RBD, washed, and released from the plate by tryptic digestion with 0.25 mg $mL^{-1}$ trypsin in the buffer containing 150 mM NaCl and 20 mM Tris-HCl pH 7.4. The selected phage particles were amplified, and the second round of selection was performed by switching the immobilizing matrix to 12 μL of MyOne Streptavidin C1 beads that were preincubated with 50 nM biotinylated RBD. Before releasing the phage particles, the binders were challenged with 5 μM non-biotinylated RBD to remove binders with fast off-rates. The third selection was repeated as the second round with 5 nM of the biotinylated RBD. After three rounds of selection, the phagemid was sub-cloned into pSb_init vector by fragment-exchange (FX) cloning and transformed into *E. coli* MC1061 for further screening at a single-colony level[24,25].

**ELISA—sybody selection.** Single colonies carrying sybody-encoding genes in the vector pSb-init were inoculated into 96-well plates. Cells were grown at 37 °C for 5 h in a shaking incubator at 300 rpm before 1:20 diluted into 1 mL of fresh TB medium supplemented with 25 μg $mL^{-1}$ chloramphenicol. Cells were induced with arabinose as mentioned earlier at 22 °C for 17 h before harvested by centrifugation at 3220 × g for 30 min. Cells were resuspended in TES Buffer (20% (w/v) sucrose, 0.5 mM EDTA, 0.5 μg/mL lysozyme, 50 mM Tris-HCl pH 8.0) and shaken for 30 min at room temperature (RT; 22–25 °C). To the lysate, 1 mL of TBS (150 mM NaCl, 20 mM Tris-HCl pH 7.4) with 1 mM $MgCl_2$ was added. The mixtures, still in the plate, were then centrifuged at 3220 × g for 30 min at 4 °C. The supernatant containing sybodies was used as directed for ELISA or FSEC assay (below).

For ELISA, protein A was incubated with Maxi-Sorp plate 96 well (Cat. 442404, Thermo Fisher) at 4 °C for 16 h. The solution was then removed and the plate was blocked by 0.5% (w/v) BSA in TBS buffer for 30 min at RT. The plate was washed three times using TBS before added with anti-Myc antibodies (Cat. M4439, Sigma) at 1:2000 dilution in TBS-BSA-T buffer (TBS supplemented with 0.5% (w/v) BSA and 0.05% (v/v) Tween 20). The antibody was allowed to bind to protein A for 20 min at RT. The plate was then washed three times with TBST (TBS supplemented with 0.05% Tween 20). Myc-tagged sybody prepared above was added and incubated for 20 min at RT. After washing three times with TBST, biotinylated RBD or MBP (the maltose-binding protein, as a control) was added to each well to a final concentration of 50 nM. After incubation for 20 min at RT, the solution was discarded and the plate was rinsed three times with TBST. Streptavidin conjugated with horseradish peroxidase (HRP) was added to each well (1:5000, Cat. S2438, Sigma). After incubation at RT for 30 min, the plate was washed three times again with TBST. ELISA signal (absorbance at 650 nm) was developed by adding 100 μL of developing buffer (51 mM $Na_2HPO_4$, 24 mM citric acid, 0.006% (v/v) $H_2O_2$, 0.1 mg $mL^{-1}$ 3,3',5,5'-tetramethylbenzidine) followed by incubation at RT.

**Sybody selection—FSEC.** To rapidly characterize RBD binders without purification, we have developed an analytic, FSEC-based assay as follows. Biotinylated $RBD_{avi}$ was bound to streptavidin (Cat. 16955, AAT Bioquest), which was fluorescently labeled by fluorescein via amine coupling. The complex is named as FL-$RBD_{avi}$. To 0.5 μM of FL-$RBD_{avi}$, cell lysate containing unpurified sybodies was added to an estimated concentration of 0.019 mg $mL^{-1}$, assuming an expression level of 20 mg $L^{-1}$. The mixture was loaded onto an analytic gel filtration column (Cat. 9F16206, Sepax) connected to an high-performance liquid chromatography (HPLC) system equipped with a fluorescence detector (RF-20A, Shimadzu). The profile was monitored by fluorescence at the excitation/emission pair of 482/508 nm. Periplasmic extract without sybodies was used as the negative control. Binders

can be identified based on earlier retention volume, presumably reflecting the bigger size of the FL-RBD$_{avi}$–sybody complex than the FL-RBD$_{avi}$ alone.

**BLI assay**. The binding kinetics was measured using a BLI assay with an Octet RED96 system (ForteBio). Biotinylated RBD was immobilized on a SA sensor (Cat. 18-5019) that was coated with streptavidin by incubating the sensor in 2 µg mL$^{-1}$ of RBD in Kinetic Buffer (0.005% (v/v) Tween 20, 150 mM NaCl, 20 mM Tris HCl pH 8.0) at 30 °C. The sensor was equilibrated (baseline) for 120 s, before incubating with sybodies at various concentrations (association) for 120 s (for MR3) or 300 s (for all the others). The concentrations for SR4 are 0, 250, 500, 1000, and 2000 nM. The concentrations for MR17 are 0, 125, 250, 500, and 1000 nM. The concentrations for MR3/MR4 are 0, 12.5, 25, 50, and 100 nM. The sensor was then moved into a sybody-free buffer for dissociation and the signal was monitored for 600 s. Data were fitted for a 1:1 stoichiometry for $K_D$, $k_{on}$, and $k_{off}$ calculations using the built-in software Data Analysis 10.0.

For the binding between MR3 and RBD mutants, MR3 was first chemically biotinylated using an amine-reactive crosslinker (Cat. 21338, Thermo Fisher). Briefly, MR3 purified in PBS buffer at 2.9 mg mL$^{-1}$ was incubated with equimolar of the crosslinker for 30 min at RT (20–22 °C). The reaction was quenched by Tris. Desalted MR3 (2 µg mL$^{-1}$) was immobilized onto an SA sensor and the BLI assays were carried out essentially as above with slight differences in association (120 or 240 s) and dissociation time (220 or 300 s) using RBD wild type and mutants as analytes at a 100-nM concentration. Binding kinetics were fitted using single-concentration curves.

For the binding between sybodies and S, sybodies were chemically labeled as for MR3 described above before immobilized on an SA sensor. BLI assays were carried out using purified S as analyte at 0, 12.5, 25, 50, and 100 nM concentrations.

For competition binding of the RBD between sybody and ACE2 (Cat. 10108-H08B, Sino Biological), the RBD was immobilized and the sensor was equilibrated as abovementioned. The sensor was then saturated using 1 µM sybody and the system was equilibrated for 180 s. After saturation, the sensor was moved into sybody solutions (50 nM) with or without 25 nM ACE2. The association of ACE2 was monitored for 600 s. As a control, the ACE2–RBD interaction was monitored using sensors without sybody incubation.

For competition binding of the RBD between MR3 and mAbs, biotinylated RBD was immobilized on an SA sensor. The sensor was saturated using 100 nM of MR3 before analyzed with 50 nM of mAbs in the presence of 50 nM of MR3. For control purposes, the binding between RBD and mAbs was carried out the same way without MR3. For competition binding of the RBD between MR3 and SR31, the assay was carried out the same way except that 100 nM of SR31 was used for pre-saturation.

For the binding assay of MR3-MR3-ABD with human serum albumin (HSA), the sensor was coated with RBD as described earlier before saturation by incubation in 200 nM MR3-MR3-ABD before soaked with 200 nM HSA for BLI signal monitoring. A control experiment was carried out in parallel but the sensor was incubated in buffer without MR3-MR3-ABD.

**Pseudotyped particle production and neutralizing assays**. The retroviral pseudotyped particles were generated by co-transfection of HEK293T cells using PEI with the expression vectors encoding the various viral envelope glycoproteins, the murine leukemia virus core/packaging components (MLV Gag-Pol), and a retroviral transfer vector harboring the gene encoding the green fluorescent protein (GFP). The S Protein expressed by phCMV-SARS-CoV and phCMV-SARS-CoV-2 has been truncated in the cytoplasmic tail by adding a stop codon, which removed 19 amino acids at the C-terminal (primers are listed under "Spike mutants" in Supplementary Data 2, along with the primers for making the D614G mutant). Supernatants that contained pseudotyped particles were harvested 48 h post-transfection and filtered through a 0.45-µm membrane before been used for neutralizing assays.

DNA encoding the human ACE2 (hACE2, NCBI accession number: NM_001371415) was cloned (from cDNA of Caco-2 cell) and constructed into pLVX-IRES-Puro lenti-vector (Addgene ID 6401). Then G glycoprotein of the vesicular stomatitis virus (VSV-G)-enveloped lentivirus pseudoparticles were packed using this vector with the human immunodeficiency virus gag pol and used to transduce VeroE6 cells. The cells were selected with 5 µg/mL puromycin. For neutralization assays, VeroE6-hACE2 cells (10$^4$ cells/well) were seeded in a 48-well plate and infected 24 h later with 100 µL of virus supernatant in a final volume of 150 µL. Sybodies were preincubated with the pseudotype samples for 1 h at 37 °C prior to cell/virus co-incubation. After 6 h of co-incubation, the supernatants were removed and the cells were incubated in a medium for 72 h at 37 °C. GFP expression was determined by fluorescence-activated cell sorting (FACS) analysis. The infectivity of pseudotyped particles incubated with sybodies was compared with the infectivity observed using pseudotyped particles and Dulbecco's modified Eagle's medium–2% fetal calf serum only and standardized to 100%. A representative flow cytometry gating strategy for the neutralization assay is included in Supplementary Fig. 13.

The characterization of VeroE6-hACE2 cells for the neutralization assay is included in Supplementary Fig. 14. Briefly, although the pseudoviruses infected VeroE6 cells, the infection rate was only at ~3% (Supplementary Fig. 14a). The overexpression of hACE2, as judged by western blot and FACS analysis

(Supplementary Fig. 14b, c), increases the infection rate by 3–10-fold (Supplementary Fig. 14a), making the assay more robust. Neutralization assays using the sybody Fc-MR3 showed similar IC$_{50}$ values between the naive and hACE2-overexpressing VeroE6 cells (Supplementary Fig. 14d), providing validation of the assay system.

Mean and standard deviation (SD, $n = 3$) were plotted for the IC$_{50}$ experiments unless stated otherwise in figure legends.

**Plaque-reduction neutralization using authentic SARS-CoV-2**. An in-house plaque-reduction neutralization test was used as a reference for this study because virus neutralization assays are the gold standard in coronavirus serology[49]. Briefly, the virus strain BetaCoV/Munich/BavPat1/2020 (400 plaque-forming units) was preincubated with serially diluted antibodies at 37 °C for 1 h before placing the mixtures on Vero-E6 cells. After incubation for 1 h and wash, cells were fixed after 2 days with 4% formaldehyde/PBS and stained with polyclonal rabbit anti-SARS-CoV-2 antibodies (Cat. 40589-T62, Sino Biological, Beijing, China; 1:1000 dilution). After a secondary peroxidase-labeled goat anti-rabbit IgG (Cat. P0448, Agilent Dako; 1:100 dilution) incubation, the foci were colored by using a precipitate forming 3,3',5,5'-tetramethylbenzidine substrate (True Blue; Kirkegaard and Perry Laboratories) and counted to measure neutralization rate.

**Hydrogen–deuterium exchange mass spectrometry**. RBD with or without MR3 bound was prepared in HEPES buffer (150 mM NaCl, 20 mM HEPES pH 7.4) at an RBD concentration of 2 mg mL$^{-1}$. In all, 3.0 µL of each sample was dispensed into a vial and diluted 9-fold with 20 mM HEPES pH 7.4, 150 mM NaCl in 99.8% D$_2$O to start the reaction on an HDX PAL system (LEAP Technologies, Carrboro, NC). HDX measurements were taken at 0, 30, 90, 300, 900, 3600, and 7200 s at 4 °C. After each time point, an aliquot of sample was transferred to a vial in a 0.5-°C chamber where the reaction was quenched by adding an equal volume of quench buffer (4 M GuHCl, 0.5 M TCEP, 0.2 M citric acid). After 30 s of quenching, the samples were loaded (flow rate of 50 µL min$^{-1}$) onto a Protease type XIII pepsin column (NovaBioAssays LLC, Woburn, MA) pre-equilibrated with 0.1% formic acid (Buffer A) in H$_2$O using an HPLC system (Thermo Dionex Ultimate 3000 NCS- 3500RS, Sunnyvale, CA). The digested peptides were trapped and desalted using a 2.1 × 5 mm Acclaim PepMap 300 C18 µ-precolumn (300 Å, 5 µm), which was connected to a 1.0 × 50 mm Thermo Hypersil Gold column C18 (175 Å, 1.9 µm). Peptides were eluted and separated by a linear gradient of Buffer B (0.1% formic acid in 80% acetonitrile) at a flow rate of 45 µL min$^{-1}$ using the nanopump of the NCS-3500RS system. Specifically, the gradient was 4–10% Buffer B 3 min, 10–30% Buffer B over 8 min, 30–90% Buffer B over 1 min followed by isocratic flow with 90% Buffer B for 1 min. The online digestion, trapping, and desalting process was performed at 4 °C and the separation process at 0.5 °C in a temperature-controlled compartment in the HDX PAL system. Data were acquired using a Thermo LTQ Orbitrap-Elite mass spectrometer (San Jose, CA) with a Thermo H-ESI II probe. For peptide identification, mass spectra were acquired in a data-dependent scan using FTMS mode in MS1 (one microscan, 100 ms max injection time, 60k resolution at 400 $m/z$) at the $m/z$ range of 300–1500 followed by 10 CID MS2 scans in the ion trap with a ±2.0 $m/z$ isolation width. Once the peptides were identified, the deuterium uptake in HDX experiments was conducted using FTMS mode in MS1. The whole HDX-MS procedure was repeated three times for each sample and for each time point.

The MS spectra were searched in PEAKS Studio X against a homemade database including target protein with a precursor mass tolerance of ≤20 ppm and MS/MS fragment ≤0.02 Da. Retention time and sequence information for each peptide were exported to Microsoft Excel. HDX-MS data analysis was carried out using HDExaminer 2.0 (Sierra Analytics Inc., Modesto, CA). The number of D taken up (D-uptake) by each peptide at each exchange time was calculated by matching the theoretical isotope distribution pattern to the observed isotope distribution pattern and plotted as a function of exchange time. Data from triplicates were analyzed using Student's $t$ test at a 95% confidence level. D-uptake, expressed as a percentage of the theoretical value, was used to generate heat maps, butterfly comparisons, and difference plots. As controls, HDX analysis was also carried out on non-deuterated and fully deuterated samples to correct back-exchange. For fully deuterated samples, the HDX-MS procedure was the same except that the lyophilized proteins were used.

**Crystallization**. Crystallization trials were set up using a Crystal Gryphon LCP robot as follows. To a two-well sitting-drop plate, 70 µL of precipitant solution was added to the reservoir. To each well, 150 nL of protein solution was added using the LCP arm of the robot. The wells were covered with 150 nL of precipitant solution using the 96-headed tips. Plates were sealed using a tape (Cat. HR4-506, Hampton Research) and placed at 20 °C in a Rocker Imager 1000 for automatic imaging.

Crystals for the SR4-RBD complex were grown in 20% (w/v) PEG 3000, 200 mM sodium chloride, and 100 mM HEPES pH 7.5. Cryo protection was achieved by adding 20% (v/v) glycerol to the mother liquor condition. Crystals for the MR17-RBD complex were grown in 20% (w/v) PEG 3350 and 0.2 M magnesium formate. Cryo protection was achieved by adding 10% (v/v) glycerol in the mother liquor condition. Crystals for the MR3-RBD complex were obtained in 9% (w/v) PEG 8000, 0.1 M HEPES pH 7.5, 8% (v/v) ethylene glycol, and 9.6% (v/v) glycerol.

20% glycerol was included for cryo cooling. Crystals for the MR4-RBD complex were grown in 10% (w/v) PEG 8000, 200 mM zinc acetate, and 100 mM MES pH 6.0. Crystals for MR17(K99Y)-RBD were grown in 0.2 M MgCl$_2$ and 20% (w/v) PEG 3350. Cryo protection was performed by adding 30% (v/v) glycerol to the reservoir condition. Crystals were cryo-protected, harvested using a MitGen loop, and flash-cooled in liquid nitrogen before X-ray diffraction data collection.

**X-ray diffraction data collection and structure determination**. X-ray diffraction data were collected at beamline BL19U1[50] at Shanghai Synchrotron Radiation Facility. Diffraction data were collected with a $50 \times 50\,\mu m$ beam on a Pilatus detector at a distance of 300–500 mm, with oscillation of 0.5–1° and a wavelength of 0.97853 Å. Data were integrated using XDS[51] and scaled and merged using Aimless[52]. The structure was solved by molecular replacement by Phaser[53] with the RBD structure from PDB 6M0J and the sybody from PDB 5M13[24] as the search model. The model was built with $2F_o–F_c$ maps in Coot[54] and refined using Phenix[55]. Structures were visualized using PyMol[56].

**Cryo-EM sample preparation and data collection for the MR3-Spike complex**. Purified SARS-CoV-2 S was incubated with MR3 sybody in a molar ratio of 1:4 on ice for 2 h before cryo-EM sample preparation. Holey carbon grids (R1.2/1.3, 200 mesh; Quantifoil) were plasma cleaned using Solarus Model 950 Advanced Plasma System (Gatan) for 30 s with 60% of $O_2$ and 40% of $H_2$. An aliquot (~2.2 μL) of the MR3-Spike complex was applied on an above-treated grid. The grid was blotted from both sides for one time with blot time of 1 s and blot force of −1 using a Vitrobot Mark IV system (Thermo Fisher Scientific) before plunged into liquid ethane cooled by liquid nitrogen.

Movie stacks (for the cryo-EM samples were collected on a Titan Krios electron microscope (Thermo Fisher Scientific) operated at an accelerating voltage of 300 kV with a nominal magnification of 22,500.

The movies were recorded on a K2 Summit direct electron detector (Gatan) operated in the super-resolution mode (yielding a pixel size of 1.02 Å after two times binning) in an automatic manner using SerialEM[57]. Each frame was exposed for 0.15 s at the dose rate of $8\,e^-\,(Å^2\cdot s)^{-1}$ and the total accumulation time was 6.45 s, leading to a total accumulated dose of $49.6\,e^-\,(Å^2)^{-1}$ on the specimen.

**Cryo-EM 3D reconstruction**. Single-particle analysis was mainly executed in Relion3.1[58] unless otherwise specified. A total of 1753 movie stacks were aligned and summed using the MotionCor2 software[59]. After contrast transfer function (CTF) parameter determination using CTFFIND4[60], particle autopicking, manual particle checking, and reference-free two-dimensional classification, 193,764 particles remained for further processing. We used part of the data to obtain an initial model. Then all particles were refined and then re-centered against this initial model. After multiple rounds of 3D classification, we obtained a MR3-Spike map from 34,243 particles. After CTF refinement and Bayesian polishing, the map was refined to 6.68 Å resolution. The map was further refined to 6.25 Å using non-uniform refinement in cryoSPARC[61]. The overall resolution was determined based on the gold-standard criterion using an FSC of 0.143.

UCSF Chimera[62] and ChimeraX[63] were used for map segmentation and figure generation.

**Structure-based design of sybody mutants to improve binding affinity**. The structure of the MR17-RBD complex was examined using Coot[54] and PyMol[56]. A panel of 19 single mutants was designed by virtual mutation using Coot[54] followed by examining the possible increase in numbers of H-bonds, salt bridges, or hydrophobic interactions. The mutations include V31F, V31I, E35F, G47A, G47F, G47W, E52F, E52M, E52Q, S53k, S53Q, H56F, H56I, H56W, H56Y, K99Y, Q103D, Q103E, and Q103Y. Plasmids encoding the mutants were generated by site-directed mutagenesis using primers listed under "MR17 mutations" in Supplementary Data 2. The mutants were purified and characterized the same way as for MR17. Because K99Y showed higher neutralization activity than the wild type, K99W was designed for the second round.

**In vivo stability of sybody in mice**. The female 7-week-old ICR mice weighing $27 \pm 1\,g$ were IP injected with PBS or sybodies MR3, MR3-MR3, MR3-MR3-ABD, or Fc-MR3 at 25 mg kg$^{-1}$ in a final volume of 100 μL in PBS. The blood samples were collected at different time points (2 days preinjection, 6 h, 12 h, 1 day, 3 days, and 6 dpi) and subjected to neutralization assay using SARS-CoV-2 pseudotypes. Mice weights were measured till 6 dpi ($n = 4$). Mice were sacrificed at 1, 3, and 6 dpi; their vital organs (heart, liver, spleen, lung, kidney, and thymus) were fixed in 4% formaldehyde at 4 °C overnight and then embedded within paraffin, solidified, and cut to 15-μm thickness using a cryotome (Leica Microsystems). Sections were stained by hematoxylin and eosin (H&E). Scale is equal to the original magnification ×100.

**Mice challenge experiments**. C57BL/6J female mice (6–8 weeks old) were treated with adenovirus serotype 5 expressing hACE2 via the intranasal route. At 5 days post-adenovirus treatment, the mice were intranasally infected with SARS-CoV-2 strain hCoV-19/China/CAS-B001/2020 (National Microbiology Data Center

NMDCN0000102-3, GISAID databases EPI_ISL_514256-7) with a high dose of $5 \times 10^5$ TCID$_{50}$ in a volume of 50 μL. After 12 h, the mice of MR3-MR3-ABD group ($n = 3$) were given 200 μL of sybody each (25 mg kg$^{-1}$ body weight) by IP injection. The infection control group ($n = 3$) was treated with PBS buffer. Three dpi, three mice were euthanized, and the lung tissues (~1/8 of the total lungs) were fixed in 4% (v/v) paraformaldehyde for histopathological analysis using H&E staining. The rest of the lungs were weighed and homogenized for RNA extraction and virus titration by qRT-PCR using a kit (Mabsky Biotech Co., Ltd.) following the manufacturer's protocols. Mean and standard error of the mean of all three individual data points were reported. The raw data were log-transformed[64] to approximate normal distribution for statistical analysis using the two-tailed, unpaired Student's t test.

**Animals and ethical statement (mice)**. The in vivo toxicity and stability study was approved by the Institutional Animal Care and Use Committee of the Institut Pasteur of Shanghai, Chinese Academy of Sciences (Animal protocol No. A2020009) and conducted in accord with all relevant ethical regulations for animal testing and research. The mice were kept in the specific pathogen-free animal facility with controlled temperature (24 °C, range: 20–26 °C), humidity (69%, range: 40–70%), and lighting conditions (12-h light/12-h dark cycle).

The live virus study was approved by the Ethics Committees of Institute of Microbiology, Chinese Academy of Sciences (SQIMCAS2020010) and was conducted in strict accordance with the recommendations provided in the Guide for the Care and Use of Laboratory Animals of the Ministry of Science and Technology of the People's Republic of China. Animals were socially housed (2–3 mice per filter top cage), placed in Class III isolator, under controlled conditions of humidity (57.5%, range: 54–61%), temperature (22 °C, range: 21–23 °C), and light (12-h light/12-h dark cycle). All experiments with live viruses and animals were performed in a biosafety level 3 laboratory and complied with the instructions of the institutional biosafety manual.

**Animals and ethical statement (hamsters)**. Animals were handled in an ABSL3 biocontainment laboratory. The research was conducted in compliance with the Dutch legislation for the protection of animals used for scientific purposes (2014, implementing EU Directive 2010/63) and other relevant regulations. The licensed establishment where this research was conducted (Erasmus MC) has an approved OLAW Assurance # A5051-01. The research was conducted under a project license from the Dutch Central Commission on Animal experiments (CCD) and the study protocol (#17-4312) was approved by the institutional Animal Welfare Body. Animals were socially housed (2–3 animals per filter top cage, (T3, Techniplast), placed in Class III isolators, under controlled conditions of humidity (55%, range: 50–60%), temperature (21 °C, range: 19–23 °C), airflow in isolator (30 m$^3$/h, range: 25–35 m$^3$/h), and light regime (12-h light/12-h dark cycles). Food and water were available ad libitum. Animals were cared for and monitored (pre- and post-infection) by qualified personnel. The animals were sedated/anesthetized for all invasive procedures.

**Animal procedures SARS-CoV-2 (hamsters)**. Female Syrian golden hamsters (*Mesocricetus auratus*; strain RjHan:AURA, purpose bred from Janvier, France) were allowed to acclimatize and aged 6 weeks at the start; for procedures, they were briefly anesthetized by chamber induction (5 L 100% $O_2$/min and 3–5% isoflurane); 6-h prior to inoculation with the virus, groups of 5 animals were treated with 2.5 mg of sybodies in 0.5 mL via the IP route.

Animals were inoculated with $10^5$ TCID$_{50}$ of SARS-CoV-2 (isolate BetaCoV/Munich/BavPat1/2020) or PBS (mock controls) in a 100 μL volume via the intranasal route. Animals were monitored for general health status and behavior daily and were weighed regularly for the duration of the study (up to 4 days post-inoculation (d.p.i.)). On 4 d.p.i., the animals were euthanized and lung and nasal turbinates were removed for virus detection and histopathology.

**Virus detection (hamsters)**. Samples from nasal turbinates and lungs were collected postmortem for virus detection by RT-qPCR and virus isolation[46]. Briefly, tissues were homogenized in viral transport medium using a Polytron PT100 tissue grinder (Kinematica). The homogenates were clarified by low-speed centrifugation and frozen and stored at −70 °C until analysis. The SARS-CoV-2 RT-qPCR was performed and quantified as TCID$_{50}$ equivalents using a standard curve of the virus stock[65]. Levels of infectious virus were determined by inoculating confluent Vero E6 cells with 10-fold serial dilutions of sample in Opti-MEM I (1×) + GlutaMAX, supplemented with penicillin (10,000 IU mL$^{-1}$) and streptomycin (10,000 IU mL$^{-1}$). At 5 d.p.i., virus positivity was assessed by reading out cytopathic effects. Infectious virus titers (TCID$_{50}$/ml) were calculated from three replicates of each sample using the Spearman–Karber method. We would note that the TCID$_{50}$ method is less sensitive (limit of detection up to 1000 TCID$_{50}$ per milliliter) than qRT-PCR, and the regular toxicity of these homogenates in cell culture introduces further variability. Therefore, while we could not detect infectious viruses, in theory, there could still be up to 1000 TCID$_{50}$ present in tissues. In addition, viral titers in nasal turbinates are even more variable due to the fact that we normalize all titers to grams of tissue. For lung tissue, we can typically take samples of ~100 mg, which then only requires us to multiply by 10. However, in the case of nasal turbinates, 10–50 mg of tissue is

typically harvested and thus increasing variability when normalizing to grams of tissue.

Mean and standard error of the mean of all five individual data points were reported. For virus titer, the raw data were log-transformed to approximate normal distribution[64] before one-way analysis of variance followed by Tukey's multiple comparisons test. Statistical analysis of the body weight data was carried out using two-way ANOVA analysis followed by Sidak's multiple comparisons test. Exact $p$ values are included in the Source data.

**Histopathology and immunohistochemistry (hamsters)**. For histological examination, lung and nasal turbinates were collected. Tissues for light microscopic examination were fixed in 10% neutral-buffered formalin, embedded in paraffin, and 3 μm sections were stained with H&E. Sections of all tissue samples were examined for SARS-CoV-2 antigen expression by immunohistochemistry[46]. Briefly, section were rehydrated, heated in a citric acid buffer (pH 6.0) for 15 min at 100 °C, and treated with 3% $H_2O_2$ to block endogenous peroxidase. Slides were washed with PBS buffer with Tween 20 and blocked with 10% goat serum for 30 min at RT. Slides were incubated with the anti-necleoprotein (SARS-CoV-2) polyclonal antibody (from rabbit, Cat. 40143-T62, Sino Biological, Chesterbrook, PA, USA) in PBS buffer (1:1000 dilution) supplemented with 0.1% BSA. After washing, the slides were incubated with goat anti-rabbit IgG conjugated with the HRP (Cat. P0448, DAKO, Agilent Technologies Netherlands, The Netherlands) (1:100 dilution) for 1 h at RT. Horseradish activity was developed for 10 min using 3-amino-9-ethylcarbazole and the sections were counterstained with hematoxylin.

For quantitative assessment of SARS-CoV-2 infection-associated inflammation in the lung, each H&E-stained section was examined for inflammation by light microscopy using a ×2.5 objective, and the area of visibly inflamed tissue as a percentage of the total area of the lung section was estimated. Quantitative assessment of virus antigen expression in the lung was performed according to the same method but using lung sections stained by immunohistochemistry for SARS-CoV-2 antigen. Sections were examined without knowledge of the identity of the hamsters. Mean and standard error of the mean of all five individual data points were reported. Statistical analysis of the pathology scores was performed using one-way ANOVA analysis followed by Tukey's multiple comparisons test. Exact $p$ values are included in the Source data.

**Reporting summary**. Further information on research design is available in the Nature Research Reporting Summary linked to this article.

## Data availability

The structure factors and coordinates are available through the protein data bank (PDB) under accession codes 7C8V (SR4-RBD), 7C8W (MR17-RBD), and 7CAN (MR17-K99Y in complex with the RBD). The cryo-EM map of MR3-Spike has been deposited to EMDB with accession ID of EMD-31328. Source data are provided with this paper.

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

## Acknowledgements

We thank the staff members of the NCPSS Large-scale Protein Preparation System and the Electron Microscopy facility for equipment maintenance and management and staff scientists at the SSRF-BL19U1 beamline at National Facility for Protein Science (Shanghai), and the staff of BSL-3 in the Institute of Microbiology, Chinese Academy of Sciences for technical support and assistance. We thank Dr. Zhipu Luo at Soochow University (China) for helpful discussions regarding data processing. This work has been supported by the Strategic Priority Research Program of CAS (XDB37020204, D. Li; XDB29010102, Y.B.), Key Program of CAS Frontier Science (QYZDB-SSW-SMC037, D. Li), CAS Facility-based Open Research Program, the National Natural Science Foundation of China (31870726, D. Li; 31870153, D. Lavillette; 32041010, Y.B.), the One Belt and One Road major project for infectious diseases (2018ZX10101004-003, J.L., G.W.), National Key R&D Program of China (2020YFC0845900, D. Lavillette), CAS president's international fellowship initiative (2020VBA0023, D. Lavillette), Natural Science Foundation of Shanghai (20ZR1466700, D. Li; 20ZR1463900, G.W.), Shanghai Municipal Science and Technology Major Project (20431900402, D. Lavillette), Science and Technology Commission of Shanghai Municipality (18DZ2210200), Funding for Construction and Operation of Zhangjiang Laboratory (II) (19DZ2260100), and the ERINHA-Advance project (funding from the European Union's Horizon 2020 Research & Innovation program, grant agreement No. 824061). This project is included in RECOVER European Union's Horizon 2020 research and innovation program under grant agreement No. 101003589. Y.B. is supported by the NSFC Outstanding Young Scholars (31822055) and Youth Innovation Promotion Association of CAS (2017122). G.W. is supported by a G4 grant from IP, FMX, and CAS.

## Author contributions

T.L., H.C., and H.Y. selected sybodies under the supervision of C.A.J.H. and M.A.S. T.L., H.C., and H.Y. purified and crystalized protein complexes with assistance from Y.L. H.Y. biochemically characterized sybodies. B.Z. and Y.Z. performed neutralization assays under the supervision of D. Lavillette. N.Z., Y.G., and Q.S. performed mice experiments under supervision of Y.B. and G.W. T.K. and N.Z. performed histopathological analysis. C.H.G.K. performed virus neutralization. W.H. collected cryo-EM data. W.H., C.L., and Yifan Wang processed cryo-EM data under supervision of Y.C. Yanxing Wang provided S protein. W.Q. collected X-ray diffraction data. J.B. helped with molecular cloning. S.-M. K. performed half-life assays in mice. J.L. and G.W. developed reagents for the neutralizing assays. M.f.v.V. and A.S.R. designed animal studies. B.R. designed and coordinated animal studies. G.W. and Y.B. developed the mice model used for in vivo studies. X.-X.T. and C.P. collected and analyzed HDX-MS data. D. Li. conceived the project, solved the structures, analyzed data, and wrote the manuscript with inputs from H.Y., T.L., H.C., B.Z., G.W., Y.B., M.A.S., M.f.v.V., B.R., Y.C., A.S.R., H.R., and D. Lavillette.

## Competing interests

The authors declare no competing interests.
