## [Peer Review File · Nature Communications]

Reviewers' Comments:

Reviewer #1:

Remarks to the Author:

Regarding: Li et al. describe the rapid development of a sybody library towards the SARS-CoV-2 RBD. The work resulted in several high-affinity binders of which one (in a dimeric-ABD construct) provided protection in a mouse model.

The paper is of course timely given the on-going pandemic but also clear and well written. The primary strengths of the work are the display of the rapidity in which efficient binders can be developed using the sybody technique, the in-vivo results and the clear display of the stabilisation of the sybody construct using an ABD which seems to hinder the rapid renal clearance often seen in nanobody therapy. Although the latter technique has been used before in other contexts I am not aware that it has been used in nanobody/sybody development towards SARS-CoV-2 before.

There are however some weaknesses that I think should be addressed:

P1. The sybodies are developed using the RBD as the target which is highly reasonable in the first steps but in the later steps of characterising the best binders the full spike context should be accounted for also in the biophysical characterisations and not only through a pseudovirus neutralisation results. I would want to see binding assays on full (soluble) variant of SARS-CoV-2 spike with the MR17 (or MR17m), SR4 and MR3 sybodies.

P2. The MX work on the MR17 (and MR17m) and SR4 is good and informative but structural information on the MR3 is sorely lacking which is very unfortunate as it is the strongest neutraliser in the pseudovirus-based experiments and is the only sybody variant tested in the mouse model. As the paper stands now the S4/MR17 binding mode is clear but the most valuable binder MR3 remains structural uncharacterised and this is a major weakness. I understand from the MS that RBD-MR3 co-crystallisation experiments have been unfruitful. However, using electron microscopy techniques (even negative stain) the authors should be able to show the binding mode of MR3. I would suggest EM to strengthen this part. Possible alternative experiments would include mutational mapping of the MR3 epitope on the RBD although they would not be equally conclusive in my opinion.

P3. The PCR results from mice challenge experiments are not clearly demonstrated in my opinion. In the graph (Fig4E), three points are denoted for both PBS and MR3-MR3-ABD but there were 6 mice challenged in the MR3-MR3-ABD experiment and 3 in the PBS experiment. Are the points denoting PCR results from individual mice? Why is then the error bar so short? The lowest point in the PBS treated group is similar to what is seen in the MR3-MR3-ABD results. I would want to see full statistics on this to be convinced with for example a full table of the raw data in the supplements and a full description on the statistical analysis of that data. I think to see enough statistical power on the effect (if present) more mice would need to be sacrificed but since this is one of the major take home messages of the paper this is important.

P4. Likewise, the histopathological examination is unconvincing in light of the limited sample size and the presentation of only two lung histopathological images in the main paper. In a revision I would want to see the relevant parts of the full lung histopathological dataset (in supplement) along with a statistically convincing analysis of the same.

Minor points:

MP1. In the extended data figure 6. SR4 and MR17 is modelled on the 'closed' conformation cryoEM structure of the SARS-CoV-2 spike. In the figure only one of the three binding opportunities on the RBDs in the trimer is displayed. If there are any clashes in the close conformation full spike context all three binding sites should be analysed for clashes and displayed. Furthermore, from just a brief analysis by this reviewer, it is clear that for SR4 and MR17 (and MR17m) there is a significant clash between the sybodies and the glycosylation on a neighbouring RBD N343 in the full spike context. This should also be discussed in the main manuscript as it may provide hints on why the SR4 and MR17 sybodies are not equally strongly neutralising compared to MR3 (putatively MR3 binds in a way to the closed spike that is not

hindered by the neighbouring N343 glycosylation).

Reviewer #2:

Remarks to the Author:

In this paper, several Sybodies against RBD of SARS-Cov-2 have been identified. The affinity, and crystal structure (of 2 in complex with the RBD) have been determined. Artificial in vitro neutralisation tests have been performed and a few next-generation constructs (bivalent, Fc fusions, tandem repeats, and linked with albumin binding domain, or affinity matured Sybody) have been produced and tested. A preliminary neutralisation test in mice with SARS-Cov-2 has been undertaken.

The methods have been explained in full detail and even sensitive experimental data (e.g. AA sequence) have been disclosed.

The novelty is perhaps not that high as similar papers with Sybodies or Nbs from immune libraries have already been published. However, with the next generation constructs and the in vivo neutralisation test the authors go slightly beyond previous experiments.

There are a few minor remarks to be considered in a revised version:

-Line 84: Eighty sequencing 'first comers' ... What is first comers???? Did they had the highest ELSIA signal? or the order they have been cherry picked from plates or running out of the sequencer as intact sequences without doubts?

-Line 95: Of note, none of the Sybodies showed neutralisation of related SARS-Cov. This is remarkable and worrying at the same time. It asks for an explanation. Since the Sybodies were selected against RBD of SARS-Cov-2 one would expect that some of the Sybodies would cross react with the RBD on SARS-Cov (since both RBD recognise to ACE2 with the epitope recognised by the neutralising Sybodies). The means that all neutralising Sybodies recognise the few amino acids that are different in the ACE binding site of the RBD of the to related SARS protein fragments. Moreover, the worrying thing is that your Sybodies are not those we want since they will no longer work due to evolutionary drift of the virus RBD. (= emergence of sybody escape variants). Shouldn't you go for broadly cross reactive Sybodies, no? instead of specific ones?

-Figure 1A (and also in other figures and text: the kinetic rate constants are denoted with small 'k' (not capital K, which is for equilibrium association/dissociation constant).

- The stability measurement (exposure to 90°C for 20 minutes of a very low concentration of Sybody) followed by fluorescent gel filtration peak maximum is not really a stability measurement of the Sybody. It is expected that at that low Sybody concentration (9 micro/mL) not much aggregation will occur. The aggregation of protein at elevated temperature is dependent on the concentration of the DENATURED protein. You are mainly looking at the capacity of reversible unfolding.

- Line 194: Supposedly 'direct fusion of the same Sybody' means a tandem repeat of the same sybody connected by a flexible linker.

Line 219 and later: Think it should be 10 fold instead of 10 folds...

- Figure 4E: please add label on y axis (viral genome copies /gram of lungs. Additional remark: the reduced viral genome load seems in lungs of Sybody treated mice seems marginal: a reduction by factor 10 on a total of 10^9 . It is difficult to understand that this minimal reduction will lead to a 'neutralisation' of the virus.

Point-to-point response to Reviewers' comments

Reviewer #1 (Remarks to the Author):

Regarding: Li et al. describe the rapid development of a sybody library towards the SARS-CoV-2 RBD. The work resulted in several high-affinity binders of which one (in a dimeric-ABD construct) provided protection in a mouse model.

The paper is of course timely given the on-going pandemic but also clear and well written. The primary strengths of the work are the display of the rapidity in which efficient binders can be developed using the sybody technique, the in-vivo results and the clear display of the stabilisation of the sybody construct using an ABD which seems to hinder the rapid renal clearance often seen in nanobody therapy. Although the latter technique has been used before in other contexts I am not aware that it has been used in nanobody/sybody development towards SARS-CoV-2 before.

- We thank the reviewer for the supportive comments as well as the constructive criticisms.

There are however some weaknesses that I think should be addressed:

P1. The sybodies are developed using the RBD as the target which is highly reasonable in the first steps but in the later steps of characterising the best binders the full spike context should be accounted for also in the biophysical characterisations and not only through a pseudovirus neutralisation results. I would want to see binding assays on full (soluble) variant of SARS-CoV-2 spike with the MR17 (or MR17m), SR4 and MR3 sybodies.

- We performed binding experiments for SR4, MR17, and MR3. The results are now in Fig. 2G (SR4-S, 6.3 nM; MR17-S, 4.2 nM), and Fig. S7A (MR3-S, 1.72 nM).

P2. The MX work on the MR17 (and MR17m) and SR4 is good and informative but structural information on the MR3 is sorely lacking which is very unfortunate as it is the strongest neutraliser in the pseudovirus-based experiments and is the only sybody variant tested in the mouse model. As the paper stands now the S4/MR17 binding mode is clear but the most valuable binder MR3 remains structural uncharacterised and this is a major weakness. I understand from the MS that RBD-MR3 co-crystallisation experiments have been unfruitful. However, using electron microscopy techniques (even negative stain) the authors should be able to show the binding mode of MR3. I would suggest EM to strengthen this part. Possible alternative experiments would include mutational mapping of the MR3 epitope on the RBD although they would not be equally conclusive in my opinion.

- We agree that the epitope information is very important.

We performed negative staining but the results were unsatisfactory probably owing to the small size of MR3.

We ended up with using four complementary methods to characterize the epitope of MR3. First we obtained a cryo-EM map of MR3-Spike at 6.25-Å resolution. Although the density does not allow accurate model building, it shows that MR3 engages at the receptor-binding motif (RBM). To confirm this, we performed hydrogen-deuterium exchange mass-spectrometer, cross-competition with structurally characterized monoclonal antibodies and nanobodies, and mutagenesis mapping. These results provided mutually corroborative evidence that MR3 targets RBD at the RBM surface.

The results are now in Fig. 3 and Fig. S7. And they are quoted as below.

The epitope of MR3 was probed using four different complementary methods. First, we performed cryo-electron microscope (cryo-EM) single-particle analysis on the MR3-S complex which had a dissociation constant of 1.7 nM (**Fig. S7A**) and obtained a map at 6.25-Å resolution (**Fig. S7B, Fig. S8**). The major class features an S trimer with two RBMs assuming the 'up' conformation (protomers A' and B') and the third one with the 'down' conformation (protomer C'). Extra densities were observed on both 'up'-RBMs, which we interpreted as MR3 engaging at the RBM region, and termed MR3-A' and MR3-B'. Interestingly, the 'down'-RBD appears to be occupied with a third MR3 molecule that clashes with the neighboring RBD and MR3 on protomer A'. This clashing may induce conformational heterogeneity, which could compromise the resolution of the cryo-EM data. Second, we performed hydrogen-deuterium exchange (HDX) mass-spectrometer to probe RBD residues that are protected from HDX by MR3 binding. Mapping the protected residues on to the RBD structure displayed a surface that overlaps with RBM and extends further down at the side (**Fig. 3A**). Third, we conducted cross-competition assays using ACE2 and structurally characterized antibodies including four monoclonal antibodies (CB6 ^{ref.37}, CV30 ^{ref.38}, REGN10987, and REGN10933 ^{ref.13}), and three sybodies (SR4, MR17, and SR31 ^{ref.39}). MR3 competed with ACE2 and all the RBM antibodies (**Fig. 3B-3H**), but not the non-RBM binder SR31 (**Fig. 3I**). Lastly, the epitope was assessed using three sets of alanine mutants (**Fig. 3J, 3K, Fig. S9**). They include mutations of 1) RBM residues (Phe456, Gln493, Gln498, Tyr505), 2) RBM-peripheral residues (Arg346, Asn354, Arg403, Asp405, Asn450, Phe490), and 3) RBM-distal residues (Val367, Lys458, Glu465). Based on the BLI binding signal, the mutants were classified as severely impaired (0-25% of wild-type), mildly affected (26-80%), and unaffected (>80%). Among the four RBM mutants, two (F456A, Y505A) displayed severe reduction in BLI binding signal (**Fig. 3J, 3K**). The other two (Q493A, Q498A) showed mild reduction in binding (**Fig. 3K, Fig. S9A**). The RBM-peripheral

group contains one severely impaired (R403A) and one mildly affected mutant (F490A) (Fig. 3J, 3K, Fig. S9A). The rest of this group and all three in the RBM-distal group displayed similar binding profile as the wild-type (Fig. 3K, Fig. S9B). It should be noted that Lys458, although being identified as a participating residue in the HDX-MS experiment (Fig. 3A), is unlikely a major contributor to the MR3-binding based on the mutagenesis results (Fig. 3K). Overall, these results provided mutually corroborative evidence that MR3, like SR4 and MR17, targets RBD at the RBM surface and neutralizes SARS-CoV-2 by competitively blocking the ACE2-RBD interaction.

Fig. 3. HDX-MS, binding, and mutagenesis studies suggest overlap between the MR3 epitope and RBM. (A) Epitope mapping using HDX-MS. RBD residues labeled with blue indicate lesser solvent exchange as an indication of protection by MR3-binding. (B) MR3 competes ACE2 for RBD binding. The RBM surface is highlighted in cyan. (C-H)

MR3 competes RBM-targeting antibodies for RBD binding. They include monoclonal antibodies CB6 ^{ref.37} (C), CV30 ^{ref.38} (D), REGN10987 ^{ref.13} (E), REGN10933 ^{ref.13} (F), and the two sybodies SR4 (G) and MR17 (H) characterized in this study. (I) The MR3-RBD binding is compatible with SR31 which targets a non-RBM surface³⁹. In C-H, the epitopes of the antibodies are highlighted in cyan except for SR4 and MR17 which are shown in Fig. 1C/1E. In B-H, binding assays were performed using RBD immobilized on a sensor. BLI profiles were recorded with (black and blue) or without (red) pre-saturation of MR3 using indicated antibodies as analyte. The assays in I was the same as in B-H except that SR31 was used for pre-saturation. (J) Three RBD mutations with severely impaired MR3 binding. The data for the rest of 10 mutants are in Fig. S9. (K) Mapping the mutagenesis data to the RBD structure. Residues are colored based on the sensitivity to alanine mutation; mutants that showed <80% binding signal compared to the wild-type are considered to be involved in binding, assuming no allosteric effects.

Fig. S7. Binding and structural analysis suggest MR3 engage S at the RBM site. (A) Binding kinetics between MR3 and S. BLI profile was recorded using immobilized MR3 and S as analyte at indicated concentrations. **(B)** The cryo-EM map of the MR3-Spike complex. A', B', and C' denote the three S protomers. MR3-A', MR3-B', and MR3-C' labels MR3-accomodating densities on Protomers A', B', and C', respectively. Two 'up'-RBDs (orange) and one 'down'-RBD (red) are appropriately labeled.

P3. The PCR results from mice challenge experiments are not clearly demonstrated in my opinion. In the graph (Fig4E), three points are denoted for both PBS and MR3-MR3-ABD but there were 6 mice challenged in the MR3-MR3-ABD experiment and 3 in the PBS experiment.

- In the Methods section we mentioned that six mice were used in each group. This was because we originally intended to assess the protection at Day 3 and Day 14. However, based on previous results from literature (*Cell* 182, 744-753; *Cell* 182 1-10), and based on the half-life of MR3-MR3-ABD, we decided to process data only until Day 3 which correspond to the best time for assessing protection for SARS-CoV-2 infection in mice.

The number has now been changed to 3. We apologize for this error.

Are the points denoting PCR results from individual mice?

- Yes

Why is then the error bar so short?

- The error bars present standard error of the mean (SEM) and are therefore shorter than standard deviation which was incorrectly indicated in the original manuscript. We apologize for the confusion caused by the error.

The lowest point in the PBS treated group is similar to what is seen in the MR3-MR3-ABD results.

- Indeed, this reflects that the protection by MR3-MR3-ABD was modest. We have now obtained data showing stronger protection in hamsters by Fc-MR3 (Fig. 6).

I would want to see full statistics on this to be convinced with for example a full table of the raw data in the supplements and a full description on the statistical analysis of that data.

- We have uploaded the raw data as supporting information. The description of the statistical analysis has now been explained in the legend of Fig. 5E:

“Lung viral loads as determined by PCR and TCID₅₀ assay from infected mice at 3 dpi. Error bars represent standard error of the mean (SEM, n=3).”

I think to see enough statistical power on the effect (if present) more mice would need to be sacrificed but since this is one of the major take home messages of the paper this is important.

- We agree with the referee on this point and the statistical power could be stronger, but due to limited resources including space issues and availability of animals, we have not been able to repeat the experiment in the facility (Beijing, China) that the original mice experiments were carried out.

Instead, we applied access to the European Research Infrastructure on Highly Pathogenic Agents and did more animal experiments using the hamster model.

The results (Fig. 6) showed that MR3-MR3-ABD again offers modest protection for the hamsters from lung damage, but Fc-MR3 offered almost full protection from SARS-CoV-2 infection in hamsters.

The hamster results are as below.

Near-competent protection of hamsters from SARS-CoV-2 by divalent MR3

Hamsters are known to be a better model than mice for COVID-19 because hamsters develop severe symptoms upon infection⁴⁹. We therefore repeated the *in vivo* virus challenge experiment in hamsters. In addition to MR3-MR3-ABD, Fc-MR3, which showed ~90% neutralization for authentic SARS-CoV-2 at 20 ng mL⁻¹ (**Fig. 5D**), was also used, along with non-neutralizing sybodies (Sb92-Sb44-ABD, and Fc-Sb2 for the ABD- and Fc-fusion, respectively) as controls (**Fig. 6**). Hamsters were administered via the intraperitoneal route with 2.5 mg of divalent sybodies 6 h before infection with 10⁵ TCID₅₀ of SARS-CoV-2 (strain BetaCoV/Munich/BavPat1/2020).

Compared to non-infected hamsters, the virus challenge caused severe weight loss (~20%) by 4 dpi. Prophylactic intraperitoneal (IP) injection of MR3-MR3-ABD reduced the weight loss by ~50%. Remarkably, despite initial weight loss in the first two days, the Fc-MR3 group regained weight to as much as the non-infected group. As controls, neither Sb44-Sb92-ABD nor Fc-Sb2 showed any protection (**Fig. 6A**). The viral RNA load in the lung was reduced by ~7 fold by MR3-MR3-ABD, compared to the Sb92-Sb44-ABD group. Consistent with the weight-loss results, the injection of Fc-MR3 reduced the RNA load by a dramatic 6Log₁₀, falling to the detection limit (**Fig. 6B**). Infectious virus was not detectable in the lungs of animals treated with Fc-MR3, or in 3 out of 5 animals treated with MR3-MR3-ABD (**Fig. 6C**). Despite the strong inhibition on the virus replication in the lung, the sybody treatment did not reduce viral RNA load in nasal turbinates, although infectious titers were below the limit of detection (**Fig. 6D, 6E**). We note, due to technical complications in accurately assessing the TCID₅₀, there are several data points that unexpectedly fall below the detection limits (**Figure 6C and 6E**), probably causing the discrepancies between the virus load (**Figure 6B and 6D**) and infectious titer. Finally, histopathologic analysis showed that MR3-MR3-ABD again offers modest protection for the hamsters from lung damage, but Fc-MR3 offers almost full protection from SARS-CoV-2 infection (**Fig. 6F, 6G**). This may be due to the fact that the Fc-fusion has much longer half-life *in vivo* than the ABD-fusion sybody (**Fig. S12B**).

Fig. 6. Potent divalent MR3 protected hamsters against weight loss and viral replication in hamsters. (A) Body weights of hamsters treated with antibodies were measured at indicated days after inoculation with SARS-CoV-2. SARS-CoV-2 viral RNA (B and C) or infectious virus (D and E) was detected in lung (B and D) and nasal turbinates (C and E). Percentage of inflamed lung tissue (F) and percentage of lung tissue expressing SARS-CoV-2 antigen (G) estimated by microscopic examination in different groups of hamsters at four days after SARS-CoV-2 inoculation. Individual

(symbols) and mean (horizontal lines) percentages are shown. The mean % of starting weight, the mean copy number or the mean infectious titer is shown, error bars represent the standard error of mean (SEM). $n = 5$. *, $p < 0.05$; ***, $p < 0.001$; ****, $p < 0.0001$.

P4. Likewise, the histopathological examination is unconvincing in light of the limited sample size and the presentation of only two lung histopathological images in the main paper. In a revision I would want to see the relevant parts of the full lung histopathological dataset (in supplement) along with a statistically convincing analysis of the same.

The histopathology data for all three mice, along with the scoring, are now included in the Fig. S12.

Fig. S12. Histopathology of lungs from infected mice at 3 dpi. (A) Histopathology of lungs from the three mice in the PBS group. **(B)** Histopathology of lungs from the three mice from the MR3-MR3-ABD group. The left panels denote an overview of the lung at 10x magnification. The right panels denote the expanded view of the black boxes in the left panels, at 100x magnification. Bars = 100 μ m. **(C)** Scoring of the data in **A** and **B**.

Minor points:

MP1. In the extended data figure 6. SR4 and MR17 is modelled on the 'closed' conformation cryoEM structure of the SARS-CoV-2 spike. In the figure only one of the three binding opportunities on the RBDs in the trimer is displayed. If there are any clashes in the close conformation full spike context all three binding sites should be analysed for clashes and displayed.

- The structure we used to dock SR4/MR17 has three identical subunits which was assumed during the cryo-EM data processing (*Cell* 181:281). Therefore in the original manuscript we only showed binding on one subunit.

We have now docked SR4/MR17 to all three subunits in Fig. S6.

Furthermore, from just a brief analysis by this reviewer, it is clear that for SR4 and MR17 (and MR17m) there is a significant clash between the sybodies and the glycosylation on a neighbouring RBD N343 in the full spike context. This should also be discussed in the main manuscript as it may provide hints on why the SR4 and MR17 sybodies are not equally strongly neutralising compared to MR3 (putatively MR3 binds in a way to the closed spike that is not hindered by the neighbouring N343 glycosylation).

- We thank the referee for this insightful observation. Indeed, both sybodies would clash with N343-linked glycans from the anticlockwise neighbor. We have now added the information to the manuscript.

In addition, we performed mutagenesis experiments to see if the glycans affect the neutralization activity of SR4/MR17. The results are in Fig. S6 and are quoted below.

... In addition, owing to their minute sizes, SR4 and MR17 could be docked into the 'close'-S with minor clashes against the Asn343-linked glycans from the anticlockwise neighboring RBD (**Fig. S6**). Such clashes may be partly responsible for the modest neutralization activity of MR17 as its neutralizing activity increased by 21 fold upon elimination of the glycosylation by alanine mutation (**Fig. S6E**). On the contrary, the N343A mutant was more resistant to SR4 than the wild-type (**Fig. S6J**), suggesting possible involvement of the glycans in SR4 binding in the context of the S trimer.

Fig. S6. SR4 and MR17 may bind to the SARS-CoV-2 S RBD in the 'closed' conformation. (A-D) Aligning the MR17-RBD with the closed conformation (PDB ID 6VXX)² of SARS-CoV-2 S. No obvious clashes were observed except for the Asn343-linked glycans (D). (E) Neutralization assay of MR17 using pseudovirus bearing the wild-type (WT) Spike or the N343A Spike. (F-J) Aligning the SR4-RBD with the closed conformation (PDB ID 6VXX)² of SARS-CoV-2 S. No obvious clashes were observed except for the Asn343-linked glycans (I). (J) Neutralization assay of SR4 using pseudovirus bearing the wild-type (WT) Spike or the N343A Spike. The three identical chains (protomers A', B' and C') of S are colored yellow, white, and pale blue. Sybodies are colored blue. Glycan chains are shown as cyan sticks. D/I show the expanded view of the area in the magenta box in B/F. The neutralization data for the WT are from Fig. 1B. IC₅₀ values ($\mu\text{g mL}^{-1}$) are shown in square brackets in E/J.

Reviewer #2 (Remarks to the Author):

In this paper, several Sybodies against RBD of SARS-Cov-2 have been identified. The affinity, and crystal structure (of 2 in complex with the RBD) have been determined. Artificial in vitro neutralisation tests have been performed and a few next-generation constructs (bivalent, Fc fusions, tandem repeats, and linked with albumin binding domain, or affinity matured Sybody) have been produced and tested. A preliminary neutralisation test in mice with SARS-Cov-2 has been undertaken.

The methods have been explained in full detail and even sensitive experimental data (e.g. AA sequence) have been disclosed.

The novelty is perhaps not that high as similar papers with Sybodies or Nbs from immune libraries have already been published. However, with the next generation constructs and the in vivo neutralisation test the authors go slightly beyond previous experiments.

- We thank the reviewer for the supportive comments to our manuscript.

There are a few minor remarks to be considered in a revised version:

-Line 84: Eighty sequencing 'first comers' ... What is first comers???? Did they had the highest ELSIA signal? or the order they have been cherry picked from plates or running out of the sequencer as intact sequences without doubts?

- We are sorry for the confusion. During the screening process, we sequenced hundreds of clones and we only tested the first 80 unique sybodies in the original manuscript.

We have now updated the manuscript with the neutralization results for all 99 sybodies (Fig. S3) and this phrase has been removed.

-Line 95: Of note, none of the Sybodies showed neutralisation of related SARS-Cov. This is remarkable and worrying at the same time. It asks for an explanation. Since the Sybodies were selected against RBD of SARS-Cov-2 one would expect that some of the Sybodies would cross react with the RBD on SARS-Cov (since both RBD recognise to ACE2 with the epitope recognised by the neutralising Sybodies). The means that all neutralising Sybodies recognise the few amino acids that are different in the ACE binding site of the RBD of the to related SARS protein fragments. Moreover, the worrying thing is that your Sybodies are not those we want since they will no longer work due to evolutionary drift of the virus RBD. (= emergence of sybody escape variants). Shouldn't you go for broadly cross reactive Sybodies, no? instead of specific ones?

- This is a very interesting topic indeed. We have added the following text in the revised manuscript for discussion.

Of note, none of the sybodies showed noticeable neutralization activities for the closely related SARS-CoV pseudovirus (**Fig. S3B**). This is partly due to the fact that antibodies recognizing three-dimensional epitopes are sensitive not only to epitope mutations, but also to allosteric mutations that affect conformational precision of the epitope. In line with this, cross-reactive antibodies against both SARS-CoVs reported so far³¹⁻³⁴ all exhibit much weaker binding towards the unintended CoV S-RBD, despite the fact that they target conserved regions. Using strategies to block mutable regions during selection may help to develop cross-reactive sybodies.

This project was initiated in early April 2020 when the pressing need was to fight the *then* current virus. Therefore we focused on the SARS-CoV-2 RBD for this work.

We agree with the reviewer that broadly reactive antibodies are very important. We have ongoing projects for broadly reactive antibodies but the strategies are very different from the current one in order to target conserved epitopes.

-Figure 1A (and also in other figures and text: the kinetic rate constants are denoted with small 'k' (not capital K, which is for equilibrium association/dissociation constant).

- Done. We thank the reviewer for catching up this error.

- The stability measurement (exposure to 90 °C for 20 minutes of a very low concentration of Sybody) followed by fluorescent gel filtration peak maximum is not really a stability measurement of the Sybody. It is expected that at that low Sybody concentration (9 micro/mL) not much aggregation will occur. The aggregation of protein at elevated temperature is dependent on the concentration of the DENATURED protein. You are mainly looking at the capacity of reversible unfolding.

- We agree with the reviewer on this point. We have removed relevant results.

- Line 194: Supposedly 'direct fusion of the same Sybody' means a tandem repeat of the same sybody connected by a flexible linker.

Yes. We have now changed the 'direct fusion' to 'tandom fusion' to make it more clear.

Line 219 and later: Think it should be 10 fold instead of 10 folds...

- Done.

- Figure 4E: please add label on y axis (viral genome copies /gram of lungs).

- Done.

Additional remark: the reduced viral genome load seems in lungs of Sybody treated mice seems marginal: a reduction by factor 10 on a total of 10^9 . It is difficult to understand that this minimal reduction will lead to a 'neutralisation' of the virus.

- The protection effect of MR3-MR3-ABD for viral infection was indeed modest.

We have now conducted hamster experiment using both the ABD and Fc version of divalent MR3. Both reduced viral load but MR3-MR3-ABD was again modestly effective. However, the treatment with Fc-MR3 markedly reduced viral load by 6Log_{10} . We have now updated this information to the revised manuscript in Fig. 6.

Reviewers' Comments:

Reviewer #1:

Remarks to the Author:

I am happy with the further work the authors performed for their present first revision of the manuscript. However, the cryo-EM work performed needs to be described in more detail, for example: glow discharge conditions, blotting conditions, how many movies were collected, what was really the calibrated pixel size used during collection? (If the 1.02 Å/pix is twice binned it doesn't really fit with the stated magnification of 22,500X).

The standard Nature group cryo-EM data table should be used and added to an updated manuscript. Furthermore, maps (and preferably data in EMPIAR) should be deposited in the EMDDB. This should also be stated (with the deposition ID) in the "data availability" section of the manuscript.

Reviewer #2:

Remarks to the Author:

The authors responded in adequate manner on the comments and critics raised by the reviewers. They even performed additional work (neutralisation of virus in hamster) as well as additional structural analysis.

Reviewer #3:

Remarks to the Author:

This is an interesting report on the generation and use of synthetic single-chain antibodies (synthetic nano bodies = sybodies) targeting the SARS-CoV-2 S protein, specifically its receptor-binding domain (RBD) infection. In an enormous effort and body of work, the authors generated and characterised 99 sybodies with respect to their neutralising activities in pseudovirus- and "real virus"-based neutralisation assays, binding affinities and avidities to RBD, and, remarkably, high-resolution structural analysis. The in-depth characterisation culminated in two different in vivo models to test the efficacy of selected sybodies against SARS-CoV-2 infection in mice and in hamsters.

The strengths of the manuscript lie in the identification and characterisation of a number of sybodies that may hold promise in prophylactic/metaphylactic and therapeutic use to prevent COVID-19. Of particular note are the multiple different approaches to identify the binding sites of sybodies on RBD and to arrive at a low Angstrom resolution of anybody-RBD interaction with implications for blockade of virus entry via the cognate receptor, human ACE2.

However, there are a number of issues that I feel need the authors' attention

1) It is unfortunate that the antibody with the best neutralising activity M3 (and its derivatives) that was also the most promising in the hamster study (Fc-M3 and, with reduced efficacy, M3-M3-ABD) refused attempts of structural analysis that was successful with other antibodies. I do realise that a concerted effort was undertaken to indirectly show M3 binding to RBD, specifically RBM as two other sybodies with slightly lower binding affinities.

2) Neutralising activities were increased by generating sybody dimers, Fc fusions, and ABD fusions. The modifications incidentally are also known to extend the half-lives of the sybodies after parenteral application. However, it is unclear (and the authors do not dwell on) why different sybody modifications result in different efficacies in the two animal models chosen (mouse and Syrian hamster). Of note the M3-M3-ABD had a very moderate effect in mice and hamsters, while Fc-M3 apparently was quite effective in the Syrian hamster model. Was the Fc-M3 tested in mice and if so, what were the results?

3) The animal experiments were performed with groups of 3 (mice) and 5 (hamsters). As there is

no description of the statistical assay used in the M&M section, it is hard to gauge what particularly the mouse results mean. In the mouse experiment, the negative control was PBS and not some irrelevant sybody. As is\, the mouse experiment only adds incrementally to the message to be conveyed here.

4) In the hamster experiment, all 5 hamsters of all groups were sacrificed on day 4 if I read the M&M and Results correctly. It would have been interesting to see the further development of weights and titres over time. Coming back to the statistical analyses, it is unfortunate that a) some lung titrations of the negative control (2 animals in the Sb92-Sb44-ABD group) were unsuccessful (for no evident reason), and b) that both sybody negative control groups had higher titres in the nose than the PBS control (about 100-fold). These massive variances are not well discussed and, given the low number of animals, put into question the titration results as whole.

Minor issues

5) it is unfortunate that the authors, while highlighting the advantages of sybody in terms of local application (intrapulmonary) in the abstract, they do not try this application route in their experimental approach in vivo.

6)The VeroE6-hACE2 cells are not referenced. It would be interesting to see neutralising activity of the sybodies and their variants in standard VeroE6 and the transgenic cells.

9) There are quite a few grammar and orthography issues. In fact, the first sentence of the abstract is awkward and should be re-phrased.

REVIEWER COMMENTS

Reviewer #1:

I am happy with the further work the authors performed for their present first revision of the manuscript. However, the cryo-EM work performed needs to be described in more detail, for example: glow discharge conditions, blotting conditions, how many movies were collected, what was really the calibrated pixel size used during collection? (If the 1.02 Å/pix is twice binned it doesn't really fit with the stated magnification of 22,500X).

We thank the reviewer for the constructive comments.

The details are as follows and are now provided in the 'Methods' section (quoted below).

...“Holey carbon grids (R1.2/1.3, 200 mesh; Quantifoil) were plasma cleaned using Solarus Model 950 Advanced Plasma System (Gatan) for 30 s with 60% of O₂ and 40% of H₂. An aliquot (~2.2µL) of the MR3-Spike complex was applied on an above-treated grid. The grid was blotted from both sides for one time with blot time of 1 s and blot force of -1 using a Vitrobot Mark IV system (Thermo Fisher Scientific) before plunged into liquid ethane cooled by liquid nitrogen.”...

...“A total of 1,753 movie stacks were aligned and summed using MotionCor2 software”...

Regarding the pixel size:

The cryo-EM data were collected on a Titan Krios with a pre-GIF K2 camera. The calibrated pixel size for magnification of 22,500X is 1.02 Å/pixel after binning 2 time for images collected under super resolution mode. Of note, the model of SARS-CoV-2 S trimer by the Veesler group (PDB ID: 6VYB) (*Cell* 2020 183:1735) fits well into our cryo-EM map, suggesting our pixel size is correct. In addition, the settings (pixel size and magnification) and the same hardware system have been used in our previous cryo-EM studies of S that yielded maps at higher resolutions (EMDB #EMD-30660, 30661, and 30703, 2.7-2.9 Å) (*Sci Adv* 2021 7: eabe5575; *Nat Commun* 2021 12:264). We also note two publications reporting similar pixel sizes (1.06 Å/pixel or 1.07 Å/pixel) for the magnification of 22,500× for data collected with pre-GIF K2 camera on Titan Krios systems (*PLoS Pathog* 2020 16:1008392; *Proc Natl Acad Sci USA* 2019 116:10366).

The standard Nature group cryo-EM data table should be used and added to an updated manuscript. Furthermore, maps (and preferably data in EMPIAR) should be deposited in the EMDB. This should also be stated (with the deposition ID) in the "data availability" section of the manuscript.

A table (Table S3, below) with cryo-EM data collection details have been added to the revised manuscript.

The cryo-EM map of MR3-Spike has been deposited to EMDB with accession ID of EMD-31328. The information has been added to the 'data availability' section of the manuscript.

Table S3. Cryo-EM data collection and reconstruction statistics.

MR3-Spike complex	
Data collection	
EM equipment	Titan Krios
Voltage (kV)	300
Detector	K2 Summit
Pixel size (Å)	1.02
Electron dose (e ⁻ /Å ²)	49.6
Exposure time (s)	6.45
Frames	43
Defocus range (µm)	-0.8 to -2.5
Reconstruction	
Softwares	Relion 3.1 & cryoSPARC
Final particles	34,243
Symmetry	C1
Final overall resolution (Å)	6.25

Reviewer #2:

The authors responded in adequate manner on the comments and critics raised by the reviewers. They even performed additional work (neutralisation of virus in hamster) as well as additional structural analysis.

-We thank the reviewer for the supportive comments.

Reviewer #3:

This is an interesting report on the generation and use of synthetic single-chain antibodies (synthetic nano bodies = sybodies) targeting the SARS-CoV-2 S protein, specifically its receptor-binding domain (RBD) infection. In an enormous effort and body of work, the authors generated and characterised 99 sybodies with respect to their neutralising activities in pseudovirus- and "real virus"-based neutralisation assays, binding affinities and avidities to RBD, and, remarkably, high-resolution structural analysis. The in-depth characterisation culminated in two different in vivo models to test the efficacy of selected sybodies against SARS-CoV-2 infection in mice and in hamsters.

The strengths of the manuscript lie in the identification and characterisation of a number of sybodies that may hold promise in prophylactic/metaphylactic and therapeutic use to prevent COVID-19. Of particular note are the multiple different approaches to identify the binding sites of sybodies on RBD and to arrive at a low Angstrom resolution of antibody-RBD interaction with implications for blockade of virus entry via the cognate receptor, human ACE2.

We thank the Reviewer for these complimentary comments and for the specific suggestions for improving the manuscript.

However, there are a number of issues that I feel need the authors' attention

1) It is unfortunate that the antibody with the best neutralising activity M3 (and its derivatives) that was also the most promising in the hamster study (Fc-M3 and, with reduced efficacy, M3-M3-ABD) refused attempts of structural analysis that was successful with other antibodies. I do realise that a concerted effort was undertaken to indirectly show M3 binding to RBD, specifically RBM as two other sybodies with slightly lower binding affinities.

We have tried multiple approaches to crystalize MR3 with RBD but unfortunately the crystals did not diffract beyond 7 Å. They include a) MR3 with and without tags, and tag positions (N- and C-terminal); b) the ‘macrobody’ version in which the maltose-binding protein (MBP) was fused to the C-terminal of MR3 as a crystallization chaperon (*eLife* 2020 9: e53683); c) co-crystallization with one of the three non-competing sybodies LR1, LR5, and SR31 (SR31-RBD crystals diffracted beyond 2.0 Å) (PDB 7D2Z, *PLoS Pathog* 2021 17: e1009328); and d) common optimization techniques such as seeding. We are still pursuing more optimization strategies such as new crystallization chaperons for structural determination. For the current study, as the reviewer also pointed out, we used four complementary methods to probe the epitope of MR3, and the results converge to the “suggested” conclusion that MR3 epitope overlaps with the RBM.

2) Neutralising activities were increased by generating sybody dimers, Fc fusions, and ABD fusions. The modifications incidentally are also known to extend the half-lives of the sybodies after parenteral application. However, it is unclear (and the authors do not dwell on) why different sybody modifications result in different efficacies in the two animal models chosen (mouse and Syrian

hamster). Of note the M3-M3-ABD had a very moderate effect in mice and hamsters, while Fc-M3 apparently was quite effective in the Syrian hamster model. Was the Fc-M3 tested in mice and if so, what were the results?

The MR3-MR3-ABD was initially selected for *in vivo* testing in mice due to it being the most potent neutralizing sybody. Following this experiment we were able to secure additional funding that allowed us to perform a second *in vivo* study in the well-established SARS-CoV-2 hamster model, which also allowed us to test multiple modifications as well as irrelevant control sybodies. From that study it was clear that the Fc-MR3 is more potent *in vivo* (in hamsters). Very little data is available on the immunology and pharmacokinetics of antibodies/nanobodies in hamsters. The efficiency differences may be due to the fact that the Fc-fusion has much longer half-life *in vivo* than the ABD-fusion sybody (Fig. S11A).

The following paragraph has been added to the revised manuscript.

“Interestingly, while the MR3-MR3-ABD was more potent *in vitro*, it only partially protected in either mice or hamsters. This may be related to its modest half-life (Fig. S11A). Consistent with this speculation, Fc-MR3 had a much longer half-life than MR3-MR3-ABD and Fc-MR3 was more efficient in protection from SARS-CoV-2 infection and disease, although we note that the stability was tested in mice (Fig. S11A) while the protection was compared in hamsters (Fig. 6). The possible difference of tissue distribution of the sybody forms, and pharmacokinetic stability modulated by the binding affinity to Fc receptors and the binding affinity to albumin from different species, may need to be investigated to explain their different *in vivo* efficiency.”

3) The animal experiments were performed with groups of 3 (mice) and 5 (hamsters). As there is no description of the statistical assay used in the M&M section, it is hard to gauge what particularly the mouse results mean.

The following sentence has been added to the Materials and Methods section:

“...Average and standard error of mean of all three individual data points were reported and the 2-tailed, unpaired student’s t-test was used to calculate p-values between the groups...”

And the *p*-values have been added in the text:

...“Compared to the control group, the lung viral titers of the sybody group were ~50-fold lower than the PBS group, when assessed at 3 days post-infection (dpi), as judged by both RNA copies and TCID₅₀ (*p*-values = 0.165 and 0.036, respectively) (Fig. 5E). In the case of lung live virus titers, the difference is statistically significant.”...

In the mouse experiment, the negative control was PBS and not some irrelevant sybody. As is\, the mouse experiment only adds incrementally to the message to be conveyed here.

We thank the reviewer for the constructive comment. While we acknowledge that the mouse experiment should have used an irrelevant sybody as a better control compared to PBS, the irrelevant sybody was not available to us at the time and we wanted to show that these nanobodies worked as a proof-of-concept before testing in hamsters.

We agree that the mice data were not as strong as the hamster data. We have modified the opening sentence of this section as the following:

...“The most potent divalent sybody (MR3-MR3) was chosen to investigate the potential of nanobodies to protect mice from SARS-CoV-2 infection in a proof-of-concept experiment.”...

In addition, the histopathology data for the mice experiment have been relocated to the supplementary information (Fig. S12).

4) In the hamster experiment, all 5 hamsters of all groups were sacrificed on day 4 if I read the M&M and Results correctly. It would have been interesting to see the further development of weights and titres over time.

From previous studies and literature, we know that SARS-CoV-2 infection in hamsters results in peak weight loss, histopathology and virus replication in respiratory tissues on day 4/5. After which the weights return to normal, and viral titers drop. Therefore, we believe that day 4 was the optimal time point to determine the protective efficacy of neutralizing sybodies.

Coming back to the statistical analyses, it is unfortunate that a) some lung titrations of the negative control (2 animals in the Sb92-Sb44-ABD group) were unsuccessful (for no evident reason),

In general, some variability in viral titers/pathology etc. is to be expected, *i.a.* as no fully inbred strain of hamsters is available. Detection of SARS-CoV-2 is most sensitive when using qRT-PCR, which is the data we base our conclusions on here. However, in the context of potential for transmission, we also performed virus titrations on homogenates of lung and nasal turbinates, to detect the presence of infectious virus. Unfortunately, this method is less sensitive and due to regular toxicity of these homogenates in cell culture increases the limit of detection up to 1000 TCID₅₀/mL. Therefore, while we could not detect infectious virus, in theory, there could still be up to 1000 TCID₅₀ present in tissues.

and b) that both sybody negative control groups had higher titres in the nose than the PBS control (about 100-fold). These massive variances are not well discussed and, given the low number of animals, put into question the titration results as whole.

In addition to the issue described above, viral titers in nasal turbinates are even more variable due to the fact that we normalize all titers to grams of tissue. For lung tissue, we can typically take samples ~100 mg, which then only requires us to multiply by 10. However, in the case of nasal turbinates, 10-50 mg of tissue is typically harvested and thus increasing variability when normalizing to g tissue.

Taking all inherent technical limitations into account, the outcomes clearly demonstrate viral replication in the organs.

Minor issues

5) it is unfortunate that the authors, while highlighting the advantages of sybody in terms of local application (intrapulmonary) in the abstract, they do not try this application route in their experimental approach in vivo.

This is indeed an experiment we have longed for. However, the BSL3 facilities have been on high demands, and the access to device allowing nebulization in BSL3 is even rarer. Therefore, we have been unable to secure time slots to test the effect of intrapulmonary administration.

6) The VeroE6-hACE2 cells are not referenced. It would be interesting to see neutralising activity of the sybodies and their variants in standard VeroE6 and the transgenic cells.

The VeroE6-hACE2 are home-made and they have been generated after lentivirus transduction and selection to express stably hACE2. In our hands, the neutralization assay on the naïve VeroE6 that we have in our laboratory is not very robust. In addition, so far, to our knowledge, no difference in neutralization has been observed between naïve cells and cells overexpressing hACE2.

9) There are quite a few grammar and orthography issues. In fact, the first sentence of the abstract is awkward and should be re-phrased.

We have gone through the manuscript carefully and corrected several grammar errors.

The first sentence has been modified as follows:

“SARS-CoV-2, the causative agent of COVID-19, features a receptor-binding domain (RBD) for binding to the host cell ACE2 protein.”

Reviewers' Comments:

Reviewer #1:

Remarks to the Author:

I am content after the changes in this version.

Reviewer #3 (Remarks to the Author):

Thank you for your rebuttal of my queries. There remain issues that need resolution or explanation and my responses are in your text as appropriate.

1) It is unfortunate that the antibody with the best neutralising activity M3 (and its derivatives) that was also the most promising in the hamster study (Fc-M3 and, with reduced efficacy, M3-M3-ABD) refused attempts of structural analysis that was successful with other antibodies. I do realise that a concerted effort was undertaken to indirectly show M3 binding to RBD, specifically RBM as two other sybodies with slightly lower binding affinities.

We have tried multiple approaches to crystalize MR3 with RBD but unfortunately the crystals did not diffract beyond 7 Å. They include *a*) MR3 with and without tags, and tag positions (N- and C-terminal); *b*) the ‘macrobody’ version in which the maltose-binding protein (MBP) was fused to the C-terminal of MR3 as a crystallization chaperon (*eLife* 2020 9: e53683); *c*) co-crystallization with one of the three non-competing sybodies LR1, LR5, and SR31 (SR31-RBD crystals diffracted beyond 2.0 Å) (PDB 7D2Z, *PLoS Pathog* 2021 17: e1009328); and *d*) common optimization techniques such as seeding. We are still pursuing more optimization strategies such as new crystallization chaperons for structural determination. For the current study, as the reviewer also pointed out, we used four complementary methods to probe the epitope of MR3, and the results converge to the “suggested” conclusion that MR3 epitope overlaps with the RBM.

Thank you for this response, which is quite reasonable.

2) Neutralising activities were increased by generating sybody dimers, Fc fusions, and ABD fusions. The modifications incidentally are also known to extend the half-lives of the sybodies after parenteral application. However, it is unclear (and the authors do not dwell on) why different sybody modifications result in different efficacies in the two animal models chosen (mouse and Syrian hamster). Of note the M3-M3-ABD had a very moderate effect in mice and hamsters, while Fc-M3 apparently was quite effective in the Syrian hamster model. Was the Fc-M3 tested in mice and if so, what were the results?

The MR3-MR3-ABD was initially selected for *in vivo* testing in mice due to it being the most potent neutralizing sybody. Following this experiment we were able to secure additional funding that allowed us to perform a second *in vivo* study in the well-established SARS-CoV-2 hamster model, which also allowed us to test multiple modifications as well as irrelevant control sybodies. From that study it was clear that the Fc-MR3 is more potent *in vivo* (in hamsters). Very little data is available on the immunology and pharmacokinetics of antibodies/nanobodies in hamsters. The efficiency differences may be due to the fact that the Fc-fusion has much longer half-life *in vivo* than the ABD-fusion sybody (Fig. S11A).

The following paragraph has been added to the revised manuscript.

“Interestingly, while the MR3-MR3-ABD was more potent *in vitro*, it only partially protected in either mice or hamsters. This may be related to its modest half-life (Fig. S11A). Consistent with this speculation, Fc-MR3 had a much longer half-life than MR3-MR3-ABD and Fc-MR3

was more efficient in protection from SARS-CoV-2 infection and disease, although we note that the stability was tested in mice (Fig. S11A) while the protection was compared in hamsters (Fig. 6). The possible difference of tissue distribution of the sybody forms, and pharmacokinetic stability modulated by the binding affinity to Fc receptors and the binding affinity to albumin from different species, may need to be investigated to explain their different *in vivo* efficiency.”

This is a reasonable paragraph to add. It would have been better, however, to assess the half-life of antibodies/sybodyes in the species in which protection studies are performed.

3) The animal experiments were performed with groups of 3 (mice) and 5 (hamsters). As there is no description of the statistical assay used in the M&M section, it is hard to gauge what particularly the mouse results mean.

The following sentence has been added to the Materials and Methods section:

“...Average and standard error of mean of all three individual data points were reported and the 2-tailed, unpaired student’s t-test was used to calculate p-values between the groups...”

And the *p*-values have been added in the text:

...“Compared to the control group, the lung viral titers of the sybody group were ~50-fold lower than the PBS group, when assessed at 3 days post-infection (dpi), as judged by both RNA copies and TCID₅₀ (*p*-values = 0.165 and 0.036, respectively) (Fig. 5E). In the case of lung live virus titers, the difference is statistically significant.”...

This remains an important issue and has not been addressed in a satisfactory fashion. In essence, arguably the most important experiment (at least from a treatment perspective) yields results that are NOT significant. The conclusion that the lung titers are significantly different is incorrect as multiple comparisons were done and there apparently was no correction for multiple comparisons

In the mouse experiment, the negative control was PBS and not some irrelevant sybody. As is\, the mouse experiment only adds incrementally to the message to be conveyed here.

We thank the reviewer for the constructive comment. While we acknowledge that the mouse experiment should have used an irrelevant sybody as a better control compared to PBS, the irrelevant sybody was not available to us at the time and we wanted to show that these nanobodies worked as a proof-of-concept before testing in hamsters.

We agree that the mice data were not as strong as the hamster data. We have modified the opening sentence of this section as the following:

...“The most potent divalent sybody (MR3-MR3) was chosen to investigate the potential of nanobodies to protect mice from SARS-CoV-2 infection in a proof-of-concept experiment.”...

In addition, the histopathology data for the mice experiment have been relocated to the supplementary information (Fig. S12).

4) In the hamster experiment, all 5 hamsters of all groups were sacrificed on day 4 if I read the M&M and Results correctly. It would have been interesting to see the further development of weights and titres over time.

From previous studies and literature, we know that SARS-CoV-2 infection in hamsters results in peak weight loss, histopathology and virus replication in respiratory tissues on day 4/5. After which the weights return to normal, and viral titers drop. Therefore, we believe that day 4 was the optimal time point to determine the protective efficacy of neutralizing sybodies.

There is, as the authors say, some lab-to-lab and experiment-to-experiment vaccination. A longitudinal study involving additional experiments would have been much more powerful and given more meaningful data.

Coming back to the statistical analyses, it is unfortunate that a) some lung titrations of the negative control (2 animals in the Sb92-Sb44-ABD group) were unsuccessful (for no evident reason),

In general, some variability in viral titers/pathology etc. is to be expected, *i.a.* as no fully inbred strain of hamsters is available. Detection of SARS-CoV-2 is most sensitive when using qRT-PCR, which is the data we base our conclusions on here. However, in the context of potential for transmission, we also performed virus titrations on homogenates of lung and nasal turbinates, to detect the presence of infectious virus. Unfortunately, this method is less sensitive and due to regular toxicity of these homogenates in cell culture increases the limit of detection up to 1000 TCID₅₀/mL. Therefore, while we could not detect infectious virus, in theory, there could still be up to 1000 TCID₅₀ present in tissues.

Thank you for the clarification

and b) that both sybody negative control groups had higher titres in the nose than the PBS control (about 100-fold). These massive variances are not well discussed and, given the low number of animals, put into question the titration results as whole.

In addition to the issue described above, viral titers in nasal turbinates are even more variable due to the fact that we normalize all titers to grams of tissue. For lung tissue, we can typically take samples ~100 mg, which then only requires us to multiply by 10. However, in the case of nasal turbinates, 10-50 mg of tissue is typically harvested and thus increasing variability when normalizing to g tissue. Taking all inherent technical limitations into account, the outcomes clearly demonstrate viral replication in the organs.

This should be included in the manuscript, rather than an explanation for the benefit of the reviewers and the editors. Again, this shows the limitation of the in vivo experiments with the low animal numbers.

Minor issues

5) it is unfortunate that the authors, while highlighting the advantages of sybody in terms of local application (intrapulmonary) in the abstract, they do not try this application route in their experimental approach in vivo.

This is indeed an experiment we have longed for. However, the BSL3 facilities have been on high demands, and the access to device allowing nebulization in BSL3 is even rarer. Therefore, we have been unable to secure time slots to test the effect of intrapulmonary administration.

As discussed above, the animal experimentation to evaluate the therapeutic potential of the sybodies is the most important aspect of this study. Therefore, the absence of BSL-3 space, while an issue of course, cannot serve as an excuse.

6) The VeroE6-hACE2 cells are not referenced. It would be interesting to see neutralising activity of the sybodies and their variants in standard VeroE6 and the transgenic cells.

The VeroE6-hACE2 are home-made and they have been generated after lentivirus transduction and selection to express stably hACE2. In our hands, the neutralization assay on the naïve VeroE6 that we have in our laboratory is not very robust. In addition, so far, to our knowledge, no difference in neutralization has been observed between naïve cells and cells overexpressing hACE2.

The generation of the cells and their characterization have to be described and the statement needs to be corroborated by references.

REVIEWERS' COMMENTS

Reviewer #1 (Remarks to the Author):

I am content after the changes in this version.

**** Thank you.**

Reviewer #3 (Remarks to the Author):

Thank you for your rebuttal of my queries. There remain issues that need resolution or explanation and my responses are in your text as appropriate.

**** We thank the reviewer once again for highlighting the remaining issues. We have taken them seriously and have addressed them as below.**

1) It is unfortunate that the antibody with the best neutralising activity M3 (and its derivatives) that was also the most promising in the hamster study (Fc-M3 and, with reduced efficacy, M3-M3-ABD) refused attempts of structural analysis that was successful with other antibodies. I do realise that a concerted effort was undertaken to indirectly show M3 binding to RBD, specifically RBM as two other sybodies with slightly lower binding affinities.

We have tried multiple approaches to crystalize MR3 with RBD but unfortunately the crystals did not diffract beyond 7 Å. They include a) MR3 with and without tags, and tag positions (N- and C-terminal); b) the ‘macrobody’ version in which the maltose-binding protein (MBP) was fused to the C-terminal of MR3 as a crystallization chaperon (*eLife* 2020 9: e53683); c) co-crystallization with one of the three non-competing sybodies LR1, LR5, and SR31 (SR31-RBD crystals diffracted beyond 2.0 Å) (PDB 7D2Z, *PLoS Pathog* 2021 17: e1009328); and d) common optimization techniques such as seeding. We are still pursuing more optimization strategies such as new crystallization chaperons for structural determination. For the current study, as the reviewer also pointed out, we used four complementary methods to probe the epitope of MR3, and the results converge to the “suggested” conclusion that MR3 epitope overlaps with the RBM.

Thank you for this response, which is quite reasonable.

**** Thanks you.**

2) Neutralising activities were increased by generating sybody dimers, Fc fusions, and ABD fusions. The modifications incidentally are also known to extend the half-lives of the sybodies after parenteral application. However, it is unclear (and the authors do not dwell on) why different sybody modifications result in different efficacies in the two animal models chosen (mouse and Syrian

hamster). Of note the M3-M3-ABD had a very moderate effect in mice and hamsters, while Fc-M3 apparently was quite effective in the Syrian hamster model. Was the Fc-M3 tested in mice and if so, what were the results?

The MR3-MR3-ABD was initially selected for *in vivo* testing in mice due to it being the most potent neutralizing sybody. Following this experiment we were able to secure additional funding that allowed us to perform a second *in vivo* study in the well-established SARS-CoV-2 hamster model, which also allowed us to test multiple modifications as well as irrelevant control sybodies. From that study it was clear that the Fc-MR3 is more potent *in vivo* (in hamsters). Very little data is available on the immunology and pharmacokinetics of antibodies/nanobodies in hamsters. The efficiency differences may be due to the fact that the Fc-fusion has much longer half-life *in vivo* than the ABD-fusion sybody (Fig. S11A).

The following paragraph has been added to the revised manuscript.

“Interestingly, while the MR3-MR3-ABD was more potent *in vitro*, it only partially protected in either mice or hamsters. This may be related to its modest half-life (Fig. S11A). Consistent with this speculation, Fc-MR3 had a much longer half-life than MR3-MR3-ABD and Fc-MR3 was more efficient in protection from SARS-CoV-2 infection and disease, although we note that the stability was tested in mice (Fig. S11A) while the protection was compared in hamsters (Fig. 6). The possible difference of tissue distribution of the sybody forms, and pharmacokinetic stability modulated by the binding affinity to Fc receptors and the binding affinity to albumin from different species, may need to be investigated to explain their different *in vivo* efficiency.”

This is a reasonable paragraph to add. It would have been better, however, to assess the half-life of antibodies/sybodies in the species in which protection studies are performed.

**** We agree with the referee on this point. Indeed, very little is known about the pharmacokinetic stability of ABD or human Fc in hamsters. This is worth investigating in a follow-up study.**

3) The animal experiments were performed with groups of 3 (mice) and 5 (hamsters). As there is no description of the statistical assay used in the M&M section, it is hard to gauge what particularly the mouse results mean.

The following sentence has been added to the Materials and Methods section:

“...Average and standard error of mean of all three individual data points were reported and the 2-tailed, unpaired student’s t-test was used to calculate p-values between the groups...”

And the *p*-values have been added in the text:

...“Compared to the control group, the lung viral titers of the sybody group were ~50-fold

lower than the PBS group, when assessed at 3 days post-infection (dpi), as judged by both RNA copies and TCID₅₀ (*p*-values = 0.165 and 0.036, respectively) (Fig. 5E). In the case of lung live virus titers, the difference is statistically significant.”...

This remains an important issue and has not been addressed in a satisfactory fashion. In essence, arguably the most important experiment (at least from a treatment perspective) yields results that are NOT significant. The conclusion that the lung titers are significantly different is incorrect as multiple comparisons were done and there apparently was no correction for multiple comparisons.

**** The statistical analysis was done separately for the qRT-PCR and TCID₅₀ data. No multiple comparisons were carried out. In the previous version, the two datasets were plotted together using two different Y-axis. We realize this may be misleading – readers may think it applies multiple comparisons. We have now plotted the data in two separate panels. We apologize for the confusion caused.**

The concern the reviewer expresses prompted us to check the statistical analysis thoroughly. As a result, we found that the previous statistical analysis could be improved.

Specifically, raw data were used for student’s *t*-test for the virus titer. Because virus replication follows an exponential distribution, statistical tests that assume normal (Gaussian distribution) would be unsuited when raw data are used. Instead, the raw data should be mathematically transformed to approximate normal distribution before analysis. This is commonly done by log-transformation (*J Clin Bioinforma* 2012 2: 5; *Proc Natl Acad Sci*, 2004 101: 8682-8686; *PLoS ONE* 2021 16: e0250516). We have re-analyzed our virus titer data for both mice and hamsters and updated them in Fig. 5, with the relevant text/legends below. We have also uploaded the statistical information in the Source Data file.

For the mice data, the re-analysis (student’s *t*-test) showed that the differences between the PBS and the MR3-MR3-ABD group are significant for the TCID₅₀ data but not for the qRT-PCR data. We have, therefore, amended the text as below to reflect these somewhat inconclusive observations.

“Virus titer analysis (Fig. 5c and 5d) and histopathology examinations (Supplementary Fig. 12) suggest a tendency of protection by MR3-MR3-ABD, although statistical conclusions could not be drawn because the differences between the two groups were significant for the TCID₅₀ results (Fig. 5d) but not for the qRT-PCR data (Fig. 5c).”

In addition, we have removed ‘protects mice from SARS-CoV-2’ or alike phrases throughout the text.

We note that the previous sections on mice and hamsters are now merged (both text and figure), and the hamster data are re-organized, as shown in Fig. 5 below. We also did the two-way ANOVA analysis on the body weight data (Fig. 5e) which shows significant effects of both Fc-MR3 and MR3-MR3-ABD compared to PBS or the control antibodies.

Fig. 5. Potent divalent MR3 protected hamsters against weight loss and viral replication in hamsters. (a) Neutralization assay of MR3-MR3-ABD using pseudovirus bearing the wild-type Spike (614D, black square) or the D614G mutant Spike (614G, red circle). Data are from one representative experiment of two independent experiments. (b) Neutralization of authentic SARS-CoV-2 by Fc-MR3 (red square) and MR3-MR3-ABD (blue circle) measured using a plaque-reduction assay. Mean \pm standard deviation are plotted ($n = 3$ independent experiments). (c, d) Lung viral loads as determined by PCR (c) and TCID₅₀ assay (d) from infected mice at 3 dpi. Error bars represent the standard error of the mean (SEM, $n=3$ biologically independent samples). Raw data were transformed to approximate normal distribution before statistical analysis using unpaired, two-tailed student's *t*-test. ns, not significant. ***, $p = 0.0009$. (e) Body weights of hamsters treated with antibodies (color-coded as indicated) were measured at indicated days after inoculation with SARS-CoV-2. Statistics were performed using two-way ANOVA followed by Sidak's multiple comparisons test. **, $p < 0.005$; ****, $p < 0.001$. Time point starting to show significance: PBS Vs Fc-MR3 and

Fc-Sb2 Vs Fc-MR3, 2 dpi; PBS Vs MR3-MR3-ABD and Sb92-Sb44-ABD Vs MR3-MR3-ABD, 4 dpi. SARS-CoV-2 viral RNA (**f** and **h**) or infectious virus (**g** and **i**) was detected in the lung (**f** and **g**) and the nasal turbinates (**h** and **i**). Percentage of inflamed lung tissue (**j**) and percentage of lung tissue expressing SARS-CoV-2 antigen (**k**) estimated by microscopic examination in different groups of hamsters at four days after SARS-CoV-2 inoculation. Individual (symbols) and mean (horizontal lines) percentages are shown. In **e-i**, the mean % of starting weight, the mean copy number, or the mean infectious titer is shown, error bars represent the standard error of the mean (SEM). $n = 5$ biologically independent samples. Statistical analyses were performed using one-way ANOVA followed by Tukey's multiple comparisons test on log-transformed data (**f - i**) or raw data (**j** and **k**). ns, not significant; *, $p < 0.05$; **, $p < 0.01$; ****, $p < 0.0001$. Source data of **a-k** and exact p -values of **c-k** are provided as a Source Data file.

In the mouse experiment, the negative control was PBS and not some irrelevant sybody. As is\, the mouse experiment only adds incrementally to the message to be conveyed here.

We thank the reviewer for the constructive comment. While we acknowledge that the mouse experiment should have used an irrelevant sybody as a better control compared to PBS, the irrelevant sybody was not available to us at the time and we wanted to show that these nanobodies worked as a proof-of-concept before testing in hamsters.

We agree that the mice data were not as strong as the hamster data. We have modified the opening sentence of this section as the following:

...“The most potent divalent sybody (MR3-MR3) was chosen to investigate the potential of nanobodies to protect mice from SARS-CoV-2 infection in a proof-of-concept experiment.”...

In addition, the histopathology data for the mice experiment have been relocated to the supplementary information (Fig. S12).

4) In the hamster experiment, all 5 hamsters of all groups were sacrificed on day 4 if I read the M&M and Results correctly. It would have been interesting to see the further development of weights and titres over time.

From previous studies and literature, we know that SARS-CoV-2 infection in hamsters results in peak weight loss, histopathology and virus replication in respiratory tissues on day 4/5. After which the weights return to normal, and viral titers drop. Therefore, we believe that day 4 was the optimal time point to determine the protective efficacy of neutralizing sybodies.

There is, as the authors say, some lab-to-lab and experiment-to-experiment vaccination. A longitudinal study involving additional experiments would have been much more powerful and

given more meaningful data.

**** We respectfully disagree with this comment from the reviewer. In our laboratory, in 6 independent challenge studies in hamsters, we consistently see peak weight loss at day 4 and 5, after which all animals start regaining weight and back at their starting weight around day 10 post-inoculation. In addition, we know from literature and our own studies, that viruses cannot be detected or at very low levels after day 5 post-inoculation. Therefore, in our opinion, a longitudinal follow-up of nanobody treated animals will not provide further evidence of efficacy against clinical signs or virus replication.**

Coming back to the statistical analyses, it is unfortunate that a) some lung titrations of the negative control (2 animals in the Sb92-Sb44-ABD group) were unsuccessful (for no evident reason),

In general, some variability in viral titers/pathology etc. is to be expected, *i.a.* as no fully inbred strain of hamsters is available. Detection of SARS-CoV-2 is most sensitive when using qRT-PCR, which is the data we base our conclusions on here. However, in the context of potential for transmission, we also performed virus titrations on homogenates of lung and nasal turbinates, to detect the presence of infectious virus. Unfortunately, this method is less sensitive and due to regular toxicity of these homogenates in cell culture increases the limit of detection up to 1000 TCID₅₀/mL. Therefore, while we could not detect infectious virus, in theory, there could still be up to 1000 TCID₅₀ present in tissues.

Thank you for the clarification

and b) that both sybody negative control groups had higher titres in the nose than the PBS control (about 100-fold). These massive variances are not well discussed and, given the low number of animals, put into question the titration results as whole.

In addition to the issue described above, viral titers in nasal turbinates are even more variable due to the fact that we normalize all titers to grams of tissue. For lung tissue, we can typically take samples ~100 mg, which then only requires us to multiply by 10. However, in the case of nasal turbinates, 10-50 mg of tissue is typically harvested and thus increasing variability when normalizing to g tissue.

Taking all inherent technical limitations into account, the outcomes clearly demonstrate viral replication in the organs.

This should be included in the manuscript, rather than an explanation for the benefit of the reviewers and the editors. Again, this shows the limitation of the *in vivo* experiments with the low animal numbers.

**** We thank the reviewer for the suggestion. The information has been added to the revised manuscript (briefly in the Results section and more detailed in the Methods section).**

We agree that we would have gained more statistical power had more samples been included in each group.

5) it is unfortunate that the authors, while highlighting the advantages of sybody in terms of local application (intrapulmonary) in the abstract, they do not try this application route in their experimental approach in vivo.

This is indeed an experiment we have longed for. However, the BSL3 facilities have been on high demands, and the access to device allowing nebulization in BSL3 is even rarer. Therefore, we have been unable to secure time slots to test the effect of intrapulmonary administration.

As discussed above, the animal experimentation to evaluate the therapeutic potential of the sybodies is the most important aspect of this study. Therefore, the absence of BSL-3 space, while an issue of course, cannot serve as an excuse.

**** We agree that the potential for developing inhaler drugs is an attractive feature of the sybodies and we hope to test this in a follow-up study. In the current study, we used the intraperitoneal (IP) route because this would allow us to compare the efficacy to other antibodies that were published at the time. In addition, the IP route is a more robust route of administration, compared to for example local application through nebulization, which is more challenging to standardize dosing and for which our facility does not have standardized protocols.**

We realize the ‘inhaled drug’ information in the Abstract and Introduction section may give readers the impression that such studies are carried out in the present study. We have now moved them to the Results & Discussion section at the very end, as follows:

“Owing to their high stability, nanobodies can survive nebulization. Of relevance to COVID-19, an inhaled nanobody drug (ALX-0171) has gone into clinical trials for the treatment of the Respiratory Syncytial Virus⁴⁶. Because of the high stable framework as originally designed²⁴, the potential of the neutralizing sybodies for the development of inhaled therapy warrants future investigations. “

6) The VeroE6-hACE2 cells are not referenced. It would be interesting to see neutralising activity of the sybodies and their variants in standard VeroE6 and the transgenic cells.

The VeroE6-hACE2 are home-made and they have been generated after lentivirus transduction and selection to express stably hACE2. In our hands, the neutralization assay on the naïve VeroE6 that we have in our laboratory is not very robust. In addition, so far, to our knowledge, no difference in neutralization has been observed between naïve cells and cells overexpressing hACE2.

The generation of the cells and their characterization have to be described and the statement needs to be corroborated by references.

**** The generation and the characterization of the VeroE6-hACE2 cells have been added to the Method section, as follows.**

“DNA encoding the human ACE2 (hACE2, NCBI accession number: NM_001371415) was cloned (from cDNA of Caco-2 cell) and constructed into pLVX-IRES-Puro lenti-vector (Addgen ID 6401). Then G glycoprotein of the vesicular stomatitis virus (VSV-G) enveloped lentivirus pseudoparticles were packed using this vector with the human immunodeficiency virus (HIV) gag pol and used to transduce VeroE6 cells. The cells were selected with 5 µg/mL puromycin.”

The characterization of VeroE6-hACE2 cells for the neutralization assay is included in Supplementary Fig. 14. Briefly, although the pseudoviruses infected VeroE6 cells, the infection rate was only at ~3% (Supplementary Fig. 14a). The overexpression of hACE2, as judged by Western blot and FACS analysis (Supplementary Fig. 14b and 14c), increases the infection rate by at 3-10 fold (Supplementary Fig. 14a), making the assay more robust. Neutralization assays using the sybody Fc-MR3 showed similar IC₅₀ values between the naïve and hACE2-overexpressing VeroE6 cells (Supplementary Fig. 14d), providing validation of the assay system.”

and the statement needs to be corroborated by references.

**** There is no consensus in the literature as to cell lines or hACE2-overexpression for neutralization assays for SARS-CoV-2 pseudoviruses or replicating viruses. Thus, naïve VeroE6 cells have been used in the literature for replicating viruses (*Nature* 2020 579:270; *Cell* 2020 182:73). For pseudotyped particles, hACE2-overexpressing HEK293T (*Nature* 2020 584:443), HT1080 (*J Exp Med* 2020 217:e20201181), HeLa (*Science* 2020 369:731), BHK-21 (*Cell* 2020 181:271), and A549 cells (*PLoS Pathog* 2021 17:e1009392) have also been reported for this application. To our best knowledge, we are the only group using VeroE6-hACE2 cells for SARS-CoV-2 infection and neutralization assays (*PLoS Pathog* 2021 17: e1009328; *Nat Commun* 2021 12:264; *Cell Discov* 2020 6:61). By our current and previous characterization, VeroE6-hACE2 joins other established cells lines as a useful tool for the neutralization assay of SARS-CoV-2 pseudoviruses.**